# Photocatalytic C(*sp³*)–C(*sp³*) cross-coupling of carboxylic acids and alkyl halides using a nickel complex and carbon nitride

Miguel M. de Vries Ibáñez [1], Luis A. Cipriano[1], Valeria Lagostina [2], Andrea Olivati [3,4], Mario Chiesa [2], Annamaria Petrozza [3], Giovanni Di Liberto [5] & Gianvito Vilé [1] ✉

Developing robust catalytic methods for constructing C(*sp³*)–C(*sp³*) bonds is critically important for synthesizing a diverse array of drug molecules. However, this type of reaction poses significant challenges from a chemical standpoint due to issues with regioselectivity, functional group tolerance and complex catalyst design. Current metallaphotoredox approaches do not provide a viable solution because they rely on expensive, toxic, and rare iridium-based photocatalysts, severely limiting their widespread application. In this study, we introduce carbon nitride nanosheets as an efficient and sustainable alternative to traditional photocatalysts. When combined with nickel, carbon nitride nanosheets facilitates the cross-coupling of alkyl halides and carboxylic acids. Our results demonstrate a broad substrate scope and highlight the recyclability of the photocatalyst. Density functional theory calculations provide molecular insights into the role of the catalytic system in facilitating photodecarboxylation and subsequent C–C bond formation. This work expands the potential of photoredox chemistry, and offers a novel method for efficient, industrially relevant light-to-chemical conversion processes.

Despite its importance for the synthesis of pharmaceuticals and agrochemicals, forming a bond between two *sp³*–hybridized carbon atoms remains one of the most challenging reactions in chemistry[1–3]. While traditional nucleophilic-electrophilic chemistry typically requires the use of strong bases, which limits functional group tolerance, often necessitates an additional reductive step to form the C(*sp³*)–C(*sp³*) product (Fig. 1a). Methods using Pd-based catalysts do not offer a fulfilling solution as they are limited by the occurrence of *β*-hydride elimination side reactions, considerably narrowing the substrate scope (Fig. 1b)[4,5]. The emergence of visible-light-mediated metallaphotoredox catalysis in synthetic organic chemistry has opened new pathways for alternative cross-coupling reactions[6–9]. This approach, driven by open-shell mechanisms, allows for mild

reaction conditions, robust functional group tolerance, and access to redox-neutral reaction routes that are not achievable with classical thermal activation[10–14]. Moreover, the use of carboxylic acids as reactive nucleophiles is particularly appealing because these molecules are readily active, inexpensive, highly abundant, and more stable than typical cross-coupling organometallic reagents[15,16]. In 2014, MacMillan and co-workers reported a photocatalytic homogeneous decarboxylative C(*sp²*)–C(*sp³*) coupling using Ni catalyst and Ir photocatalyst[17]. Further applications of this procedure led to the construction of C(*sp³*)–C(*sp³*) bonds using similar catalytic system (Fig. 1c)[18,19]. Although this approach significantly broadens the horizons for synthetic pathways, it relies on rare and expensive photocatalysts containing Ir or Ru (*e.g.*, Ir[dF(CF₃)ppy]₂(dtbbpy)PF₆ or

[1]Department of Chemistry, Materials, and Chemical Engineering "Giulio Natta", Politecnico di Milano, Milano, Italy. [2]Department of Chemistry, University of Torino, Torino, Italy. [3]Center for Nanoscience and Technology, Istituto Italiano di Tecnologia, Milano, Italy. [4]Physics Department, Politecnico di Milano, Milano, Italy. [5]Department of Materials Science, University of Milano Bicocca, Milano, Italy. ✉e-mail: gianvito.vile@polimi.it

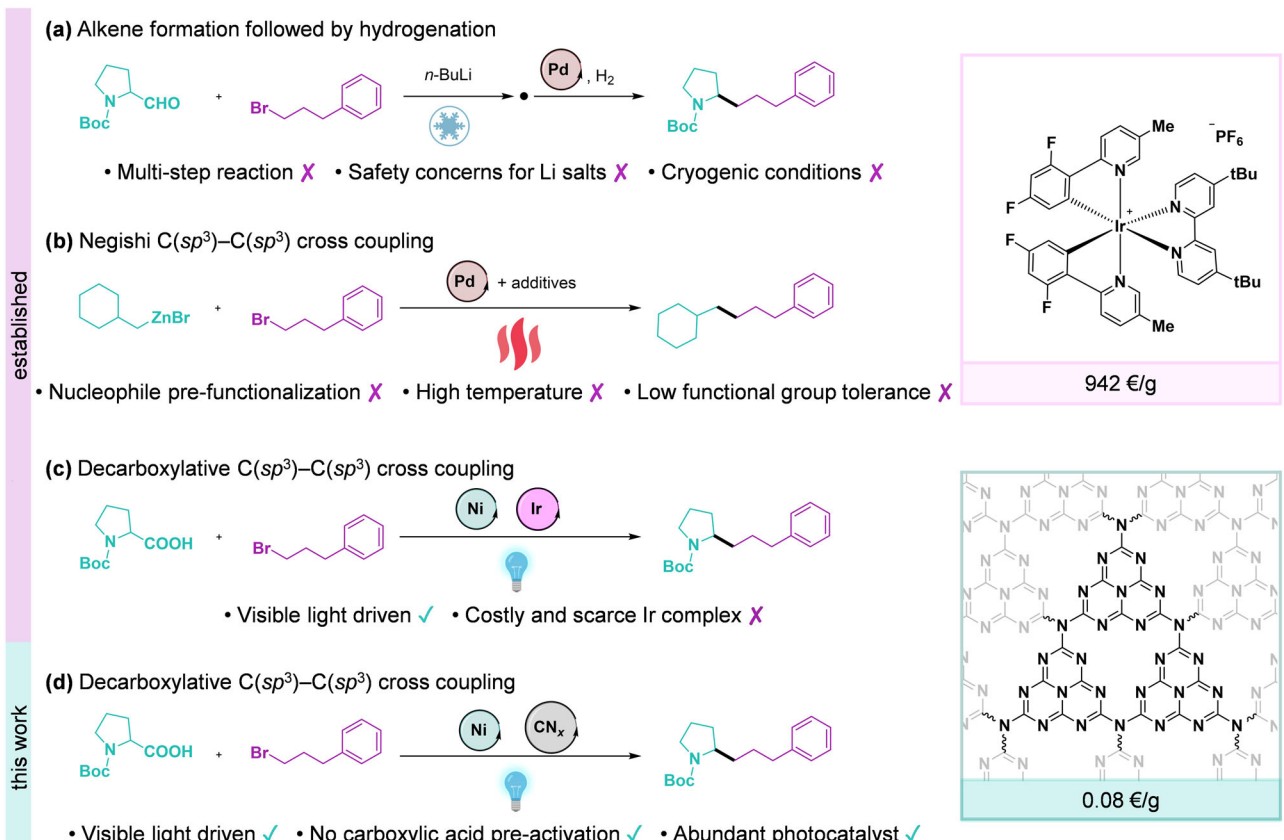

**Fig. 1 | Established and new catalytic methods for the formation of the C($sp^3$)−C($sp^3$) cross-coupled products.** Multi-step synthesis involving *n*-BuLi and Pd-catalyzed hydrogenation (**a**), Pd-catalyzed coupling of an organozinc reagent with an alkyl bromide (**b**), Ni and Ir dual catalysis under visible light (**c**), light-driven Ni catalyst with a highly abundant carbon nitride photocatalyst (**d**). The study herein points to the first semi-heterogeneous method to synthesize $sp^3$-hybridized molecules using visible light and only earth-abundant catalysts. Photocatalyst prices reported in the figure are obtained from Merck.

Ru(bpy)$_3$(PF$_6$)$_2$[20–22]. However, the scarcity of these metals[23], their toxicity[24], and high cost[25] limit the potential for large-scale application. Substituting these precious metals with cheaper, noble-metal-free, and recyclable catalysts could significantly boost the industrial viability of these routes.

In this context, semiconductors have garnered significant interest in photocatalysis due to their low cost, robustness, and ability to be filtered and reused after reaction[26–28]. Suitable semiconductor photocatalysts have a band gap that can be excited with UV or visible light to participate in single-electron transfer (SET) redox processes, which can be eventually further enhanced by doping these photocatalysts with additional elements[29–31]. Recent studies have shown that metal-free carbon nitrides (CN$_x$) are effective photocatalysts in organic synthesis and can facilitate various transformations[32–35]. Significantly, it has been demonstrated that carboxylic acid coupling partners can undergo efficient oxidatively induced radical decarboxylation to generate alkyl radical intermediates[36]. However, attempts to use carboxylic acids directly with CN$_x$ photocatalysts for C−C coupling reactions, beyond the more straightforward C($sp^2$)−C($sp^3$) bond formation[37], have been hindered by the tendency of deprotonated carboxylic complexes to form esters. Consequently, the formation of C($sp^3$)−C($sp^3$) bond has remained challenging to demonstrate.

In this work, we present the combination of carbon nitride and nickel complexes for the C($sp^3$)−C($sp^3$) cross-coupling reactions. Our approach does not employ iridium or ruthenium and does not need any pre-activation of the carboxylic acid with phthalimides, thus merging photo- and Ni-catalytic cycles in an easier manner and offering significant improvements in sustainability, catalyst reuse, and

recyclability compared to previous studies. The catalytic tests in optimized conditions highlighted the excellent performance of our system and the broad substrate tolerance of the developed dual protocol. The results are supported by Stern-Volmer experiments, electron paramagnetic resonance (EPR) analysis, and density functional theory (DFT) calculations, revealing key structural and mechanistic insights into the behavior of the catalytic system under visible light irradiation.

## Results and discussion

The synthesis of carbon nitride nanosheets (nCN$_x$) was carried out through the calcination of melamine[38]. Specifically, melamine (4 g; Sigma Aldrich, 99%) was heated at 550 °C for 3 h (heating ramp: 10 °C min$^{-1}$) in an alumina crucible to obtain graphitic carbon nitride (gCN$_x$). The obtained sample was then subjected to a thermal exfoliation step at 550 °C for 3 h (heating ramp: 2 °C min$^{-1}$) to obtain nCN$_x$ (Fig. 2a). The specific surface area of the prepared material was determined *via* N$_2$ physisorption experiments. During this step, layers within the gCN$_x$ structure are separated, creating a more porous structure. This increased surface area provides more active sites for photocatalytic reactions, thereby enhancing the overall photocatalytic activity of the nCN$_x$ photocatalyst. BET adsorption isotherms demonstrated the more porous nature of nCN$_x$ material compared to the initial gCN$_x$, which exhibited a larger surface area (23 m$^2$ g$^{-1}$) compared to gCN$_x$ (6 m$^2$ g$^{-1}$) (Fig. 2b). X-ray diffraction (XRD) was conducted to define the orientation, crystallinity and purity of the material. The diffractograms revealed two characteristic diffraction peaks at $2\theta = 13°$ and 27° (Fig. 2c), corresponding, respectively, to the (100) and (002) planes of carbon nitride. The

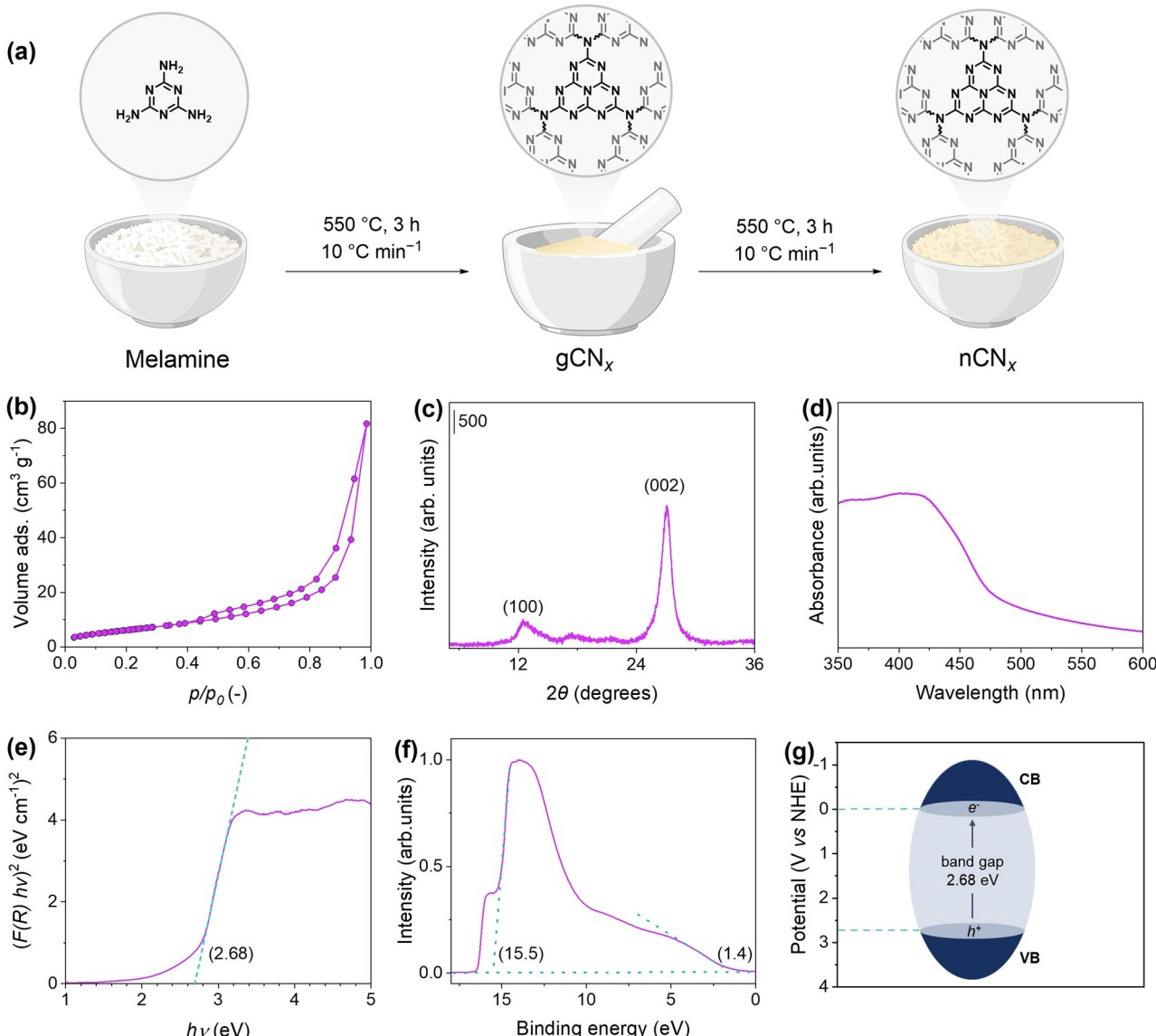

**Fig. 2 | Synthesis and characterization of nCN$_x$ photocatalyst.** Synthesis of nCN$_x$ through calcination of melamine (**a**) and characterization of the obtained nCN$_x$ sample via N$_2$ physisorption (**b**), X-ray diffraction (**c**), UV-Vis absorbance spectroscopy (**d**), corresponding Tauc plot (**e**), UPS spectra (**f**) and conductive and valance energy levels of nCN$_x$ (**g**).

former denotes the trigonal *N*-linkage of the triazine moiety, while the latter represents the stacking of aromatic rings in nCN$_x$. No other peaks were observed, confirming the absence of any crystalline impurity in the sample. Elemental analysis conducted through combustion revealed C/N ratios ranging from 0.61 to 0.67, close to the ideal value of 0.7 for the basic heptazine structure of nCN$_x$. Discrepancies from the ideal value are typically ascribed to defects arising from the thermal polymerization process. Nevertheless, the low H content confirmed a high degree of polymerization of the melamine precursor, with only a small percentage of non-polymerized units containing residual protons.

To evaluate the photophysical properties of nCN$_x$ and calculate its corresponding band gap value, ultraviolet-visible (UV-Vis) diffuse reflectance spectroscopy (DRS) measurements we have garnered significant interest re performed. DRS enabled the direct analysis of solids or powders without the need for their dissolution, making it particularly useful for studying materials like nCN$_x$, which are insoluble or poorly soluble in common solvents. Thus, this method allows for analysis of photocatalysts in their "native" form, avoiding potential

artifacts introduced by scattering processes typical of dispersions of insoluble materials. The UV-Vis spectrum of nCN$_x$ (Fig. 2d) exhibit an absorbance onset from approximately 460 nm, with a maximum around 420–425 nm, indicating effective visible light absorption and its possible utilization in photocatalytic applications. The band gap value of nCN$_x$ was determined from the Kubelka-Munk function and Tauc plot obtained from the UV-Vis spectra (Fig. 2e)[39], and was 2.68 eV. Ultraviolet photoelectron spectroscopy (UPS) was employed to elucidate the electronic band structure of nCN$_x$ (Fig. 2f). The valence energy ($E_v$) was determined to be 7.12 eV, by subtracting the width of the He I UPS spectra from the excitation energy (21.22 eV). Considering the calculated band gap energy, the conduction band energy ($E_c$) was estimated at 4.44 eV. The $E_g$, $E_v$, and $E_c$ values were further converted to electrochemical energy potentials in volts according to the reference standard [−4.44 eV *vs.* vacuum level equals 0 V *vs.* normal hydrogen electrode (NHE)][40]. Resulting in $E_v$ and $E_c$ = 2.68 and 0 eV *vs.* NHE (Fig. 2g).

Upon thoroughly characterizing the photocatalyst, we proceeded to explore its application in the decarboxylative C($sp^3$)−C($sp^3$) cross-

coupling reactions. For the optimization of the reaction conditions (Fig. S1), our initial focus was on identifying the most effective base to facilitate this transformation. In the first set of experiments, we evaluated a variety of bases to determine their impact on the yield of the desired product (compound **3a**) and the byproduct (**4a**) (Table S1). We began with cesium carbonate ($Cs_2CO_3$), which yielded only 8% of the desired product **3a** and a significantly higher 92% of byproduct **4a**. Potassium carbonate ($K_2CO_3$) improved the yield of **3a** to 34% with a concomitant reduction in **4a** to 44%. Sodium carbonate ($Na_2CO_3$) produced a similar yield for compound **3a** (33%), but a lower yield for compound **4a** (17%). Further testing of bases such as sodium bicarbonate ($NaHCO_3$) and lithium carbonate ($Li_2CO_3$) yielded negligible results for compound **3a**, with $NaHCO_3$ producing a minimal 6% of compound **4a** and $Li_2CO_3$ showing no significant activity because of its poor solubility in the reaction media. Potassium hydrogen phosphate ($K_2HPO_4$) provided moderate results with 28% yield for compound **3a** and 40% for compound **4a**. Sodium hydroxide (NaOH) and triethylamine also showed low yields for both products, with NaOH yielding no desired product and 15% of the byproduct, while triethylamine yielded 6% and 25%, respectively. The use of 1,8-diazabicyclo[5.4.0]undec-7-ene (DBU) resulted in no detectable yield of the desired product and an overwhelming 99% of the byproduct. In contrast, quinuclidine provided no yield of the desired product but a high yield of 96% for the byproduct. Other amines, such as *N-tert*-butylisopropylamine, *N,N*-diisopropylethylamine (DIPEA), and dibutylamine produced low to moderate yields, with the highest yield for the desired product being 23% from disiopropylamine. Overall, the experimental campaign clearly demonstrated that, in acetonitrile, inorganic bases worked better than organic bases due to their lower steric hindrance, higher availability of free base species in solution, and simpler interaction dynamics. These factors collectively facilitate more efficient deprotonation and base-catalyzed steps, leading to higher selectivity and yields of the desired product while minimizing the formation of byproducts. Next, we optimized the amount of base ($Na_2CO_3$) used in the reaction (Table S2). Using 1.0 equivalent of $Na_2CO_3$ yielded 27% of compound **3a** and 24% of compound **4a**. Increasing the amount to 2.0 equivalents slightly improved the yield of compound **3a** to 33%, but reduced compound **4a** to 17%. Further increases to 3.0 and 4.0 equivalents did not significantly enhance the yields. Although no significant influence on the reaction outcome was observed, 2.0 equivalents of $Na_2CO_3$ were selected as optimal for the reaction.

Subsequent experiments focused on optimizing the photocatalyst type (Fig. S2) and concentration (Table S3). Using 10 mg mL$^{-1}$ of nCN$_x$ resulted in 33% yield of compound **3a** and 17% of compound **4a**. Increasing the concentration to 12.5 mg mL$^{-1}$ of nCN$_x$ improved the yield to 41% for compound **3a** and 23% for compound **4a**. However, further increasing the concentration to 15 mg mL$^{-1}$ resulted in diminished yields. Mesoporous graphitic carbon nitride (mpgCN$_x$) was also tested at 5 mg mL$^{-1}$ yielding 20% of compound **3a** and 12% of compound **4a**, while 7.5 mg mL$^{-1}$ of mpgCN$_x$ provided better results with 34% yield of compound **3a** and 18% of compound **4a**. Higher concentration of mpgCN$_x$ (10 mg mL$^{-1}$) did not improve yields significantly. This result was likely due to transport phenomena at the reactor level. In fact, mpgCN$_x$ was more challenging to handle during the experimental campaign, as the high surface area of this material resulted in its adhesion to the vial walls, which potentially led to less efficient light irradiation. While the reaction outcome of nCN$_x$ and mpgCN$_x$ is comparable, nCN$_x$ was chosen to proceed with optimization due to its easier handling. Finally, using a potassium-doped carbon nitride, potassium poly-heptazine imide (K-PHI), (10 mg mL$^{-1}$) yielded 28% of compound **3a** and a notable 43% of compound **4a**. We then explored the effect of different wavelengths of light on the reaction (Table S4). White LEDs resulted in 6% yield of compound **3a** and 51% of compound **4a**. Green LEDs (500 nm) were ineffective for the desired product, yielding 11% of the byproduct. Blue LEDs (460 nm)

significantly improved the yield of compound **3a** to 41% and 23% for compound **4a**. Purple LEDs (420 nm) provided the best results with 54% yield of compound **3a** and 20% of compound **4a**, indicating the critical role of light wavelength in the reaction efficiency. The choice of purple LEDs as the most effective light source could be attributed to the interaction between the light wavelength and the nCN$_x$ catalyst used in the reaction. Figure 2d demonstrated that the nCN$_x$ catalyst exhibits an absorption band in the blue region of the spectrum, with a major absorbance peak around 420 nm, which corresponds to the wavelength of purple LEDs utilized. We progressed investigating the effect of varying the equivalents of carboxylic acid (Table S5). Using 1.0 equivalent resulted in 31% yield of compound **3a** and 17% of compound **4a**. Increasing the amount to 1.5 equivalents significantly improved the yield to 54% for compound **3a** and 20% for compound **4a**. Further increases to 3.0 and 5.0 equivalents resulted in lower yields, suggesting that 1.5 equivalents of carboxylic acid were optimal.

The source and amount of nickel catalyst were then optimized (Table S6). Using 0.1 equivalent of NiCl$_2$·glyme with 4,4′-dimethoxy-2,2′-bipyridine (dMeObpy) ligand yielded 54% of compound **3a** and 20% of compound **4a**. NiBr$_2$·glyme with the same ligand resulted in significantly lower yields. NiCl$_2$ with dMeObpy yielded 17% of compound **3a** and 14% of compound **4a**, while Ni(Ac)$_2$·4H$_2$O with dMeObpy provided 41% and 10% yields, respectively. NiO with dMeObpy was largely ineffective. Using 4,4′-di-*tert*-butyl-2,2′-bipyridine (dtbbpy) ligand with NiCl$_2$·glyme gave 45% of compound **3a** and 17% of compound **4a**. Increasing the amount of NiCl$_2$·glyme with dMeObpy to 0.2 equivalents resulted in the highest yield of 60% for compound **3a** but lower yield of 10% for compound **4a**. We then optimized the concentration of the reaction mixture (Table S7). Using MeCN at 0.2 M concentration yielded 30% of compound **3a** and 27% of compound **4a**. Reducing the concentration to 0.1 M significantly increased the yield to 60% for compound **3a** and reduced compound **4a** to 10%. Further reduction to 0.05 M resulted in the highest yield of 75% for compound **3a**, with no detectable byproduct.

Finally, control experiments were conducted to validate the necessity of each component in the reaction (Table 1). Under the optimized conditions, the reaction yielded 75% of **3a** with no detectable byproduct (Table 1, entry 1). Omitting the photocatalyst (nCN$_x$) resulted in no desired product and 12% of compound **4a** (Table 1, entry 2). Removing the nickel catalyst led to no yield of compound **3a** and 6% of compound **4a** (Table 1, entry 3). Without the ligand, the yield of compound **3a** dropped to 21%, and only 9% of compound **4a** was formed (Table 1, entry 4). Omitting the base entirely resulted in no reaction (Table 1, entry 5). Performing the reaction in the absence of light yielded no products, confirming the essential role of light in the photocatalytic process (Table 1, entry 6). Lastly, to further study the performance of our semi-heterogeneous system, nCN$_x$ was replaced with the benchmark homogeneous photocatalyst Ir[dF(CF$_3$) ppy]$_2$(dtbbpy)PF$_6$, widely reported in decarboxylative cross-coupling reactions[17,18]. The Ir complex under our (non-degassed) optimized conditions delivered **3a** in 61% yield (Table 1, entry 7). The diminished activity observed is likely due to quenching of the Ir complex's long-lived excited state by dissolved oxygen in solution. It must be emphasized that while a direct comparison between homogeneous and semi-heterogeneous systems under identical conditions is inherently limited due to fundamental differences in terms of active site accessibility, mass and heat transfer, dispersion, and reaction dynamics, the experiment nonetheless highlights the practical advantages of using nCN$_x$ as a robust, oxygen-tolerant photocatalyst. Its solid-state nature enables operation without the need for inert atmospheres, which simplifies handling and operational protocols. Overall, through this extensive optimization, we established a highly efficient and reliable method for C($sp^3$)−C($sp^3$) coupling using carbon nitride and nickel catalysis, optimizing the reaction from an initial yield of 8% for compound **3a** (Table S1, entry 1) to 75% (Table 1, entry 1). With

**Table 1 | Optimization of the C(*sp*³)–C(*sp*³) cross coupling reaction with corresponding control experiments**[a]

| Entry | Conditions | Yield of 3a | Yield of 4a |
|---|---|---|---|
| 1 | As shown above | 75% | n.d.[b] |
| 2 | No nCN$_x$ | n.d. | 12% |
| 3 | No Ni | n.d. | 6% |
| 4 | No ligand | 21% | 9% |
| 5 | No base | n.d. | n.d. |
| 6 | No light | n.d. | n.d. |
| 7 | Using Ir[dF(CF₃)ppy]₂(dtbbpy)PF₆ instead of nCN$_x$[c] | 61% | 5% |

[a]The reaction scheme shows standard conditions, while control experiments for the C(*sp*³)–C(*sp*³) cross-coupling reaction are included in the table. dMeObpy refers to the 4,4'-dimethoxy-2,2'-bipyridine ligand. Yields were calculated by HPLC analysis using the calibration curve of **3a**. Additional experiments are included in the Supplementary Material.
[b]Not detected.
[c]This reaction was performed with identical experimental conditions as in the reaction scheme, but with 0.02 equiv. of Ir[dF(CF₃)ppy]₂(dtbbpy)PF₆ instead of nCN$_x$.

optimal metallaphotoredox conditions in hand, we probed the generality of this method, testing differently decorated carboxylic acids and alkyl halides (Fig. 3). The model reaction was carried out using the corresponding alkyl bromide, affording the product **3a** in excellent isolated yields (71%). One of the key limitations of photocatalysis and photochemistry is the challenge of scalability due to light absorption[41]. This issue arises from the attenuation of photon flux as described by the Lambert–Beer law[42], which hinders the classical scale-up of batch reactions by simple enlargement of reaction volume. To address this, a numbering-up strategy was first employed to demonstrate reproducibility and robustness across multiple reactions. Numbering up refers, in fact, to a modular approach aimed at enhancing throughput by operating multiple identical reactions in parallel, while maintaining consistent reaction conditions. In photochemistry, this approach ensures uniform light distribution across all reactors, avoiding the challenges of light penetration and inefficiencies associated with large-scale systems[43,44]. In this context, a 3 mmol scale reaction was prepared and divided into 15 vials, obtaining 77% yield of **3a**. The result provides a proof-of-concept that emphasizes the critical role of maximizing the light-exposed surface area, particularly in heterogeneous systems where light penetration is further hindered by solid components. However, this should not be interpreted as a comprehensive scale-up. For this reason, we also did a scale-up experiment using a sizing-up approach. The reaction was scaled up by a factor of 10, from 0.2 mmol to 2 mmol, in a single round-bottom flask illuminated with four Kessil lamps (427 nm), yielding product **3a** in 63% isolated yield. This yield is lower than the 77% yield obtained using the numbering-up method, and we attribute this drop primarily to differences in the geometry of the reaction vessels, which affected the effective irradiated surface area: in the numbering-up setup, small vials provided an irradiated surface area of approximately 2.16 cm² mL⁻¹, whereas the round-bottom flask used in the scale-up experiment had a slightly lower irradiated surface area of about 1.57 cm² mL⁻¹. These results help clarify the relationship between reactor geometry, light exposure, and reaction efficiency in batch photochemistry[45], and demonstrate a ten-fold scale up is feasible with only a modest loss in yield.

Good to excellent isolated yields were obtained when expanding or reducing the proline *N*-bearing ring in substrates **3b** (64% yield) and **3c** (56% yield), which was particularly relevant given that these modifications introduced varying degrees of ring strain. A similar outcome was observed when using linear *α*-amino acids, with yields of 72% for

**3d**, 67% for **3e**, and 48% for **3f**. The lower yield of **3f** can be attributed to the formation of a radical in the benzylic position that resulted in a competitive E-2 elimination side reaction, leading to the inactivation of the substrate by the olefin formation. The functionalization of lysine presents a critical opportunity for drug discovery research because lysine residues are abundant on protein surfaces and play key roles in various biological processes, including enzymatic catalysis, protein-protein interactions, and post-translational modifications. In this context, the alkylation of *N*-protected lysine, achieved with a 74% yield (**3g**), offers a highly efficient method that could pave the path for the development of novel pharmaceutical compounds, providing a significant competitive tool in medicinal chemistry[46]. Bicyclic (**3h**) and tricyclic (**3i**) *α*-amino acids were also tested, yielding comparable results (66% and 77%, respectively). Furthermore, substituted *α*-amino acids were tested with electrophilic and nucleophilic functional groups, and all functioned as efficient coupling partners (substrates **3j** and **3k**, 49–78% yield), thus confirming the importance and optimal reactivity of heteroatom-containing cyclic carboxylic acids, independent of the substitution pattern on the cyclic moiety. The Boc-protecting group, traditionally used to shield the amino functionality during synthesis, could be substituted with the Cbz (carbobenzyloxy) protecting group (**3l**), leading to the formation of the coupled C–C product with a yield of 68%. Different alkyl bromides were then also tested. Primary alkyl bromides bearing electron-withdrawing (**3m** and **3n**) and electron-donating moieties (**3o**) could be successfully coupled (96%, 98% and 61% yield, respectively). Moreover, alkyl bromides containing nucleophilic (**3p**, 90% yield and **3q**, 64% yield) and electrophilic functional groups (**3r**, 88%) were well tolerated. Increasing the steric effect on the alkyl bromide reduced the reaction outcome (**3s**, 37%), potentially due to a more challenging oxidative addition step. The reaction outcome was reproducible by varying the electronics of secondary alkyl bromides. Oxygen-containing cyclic alkyl bromides were successfully functionalized, yielding **3t** (79%) and **3u** (75%), while aliphatic cyclic alkyl bromides also gave good results, with **3v**, **3w**, and **3x** obtained in 67%, 60%, and 59% yield, respectively. To complete the reaction scope, a levodopa precursor was successfully functionalized, leading to **3y** in 71% yield. This outcome demonstrates the potential of carbon nitride for creating high-value derivatives of key therapeutic molecules, as the ability to efficiently modify levodopa (a crucial drug in the treatment of Parkinson's disease) opens the door to the development of novel therapeutic agents, offering substantial

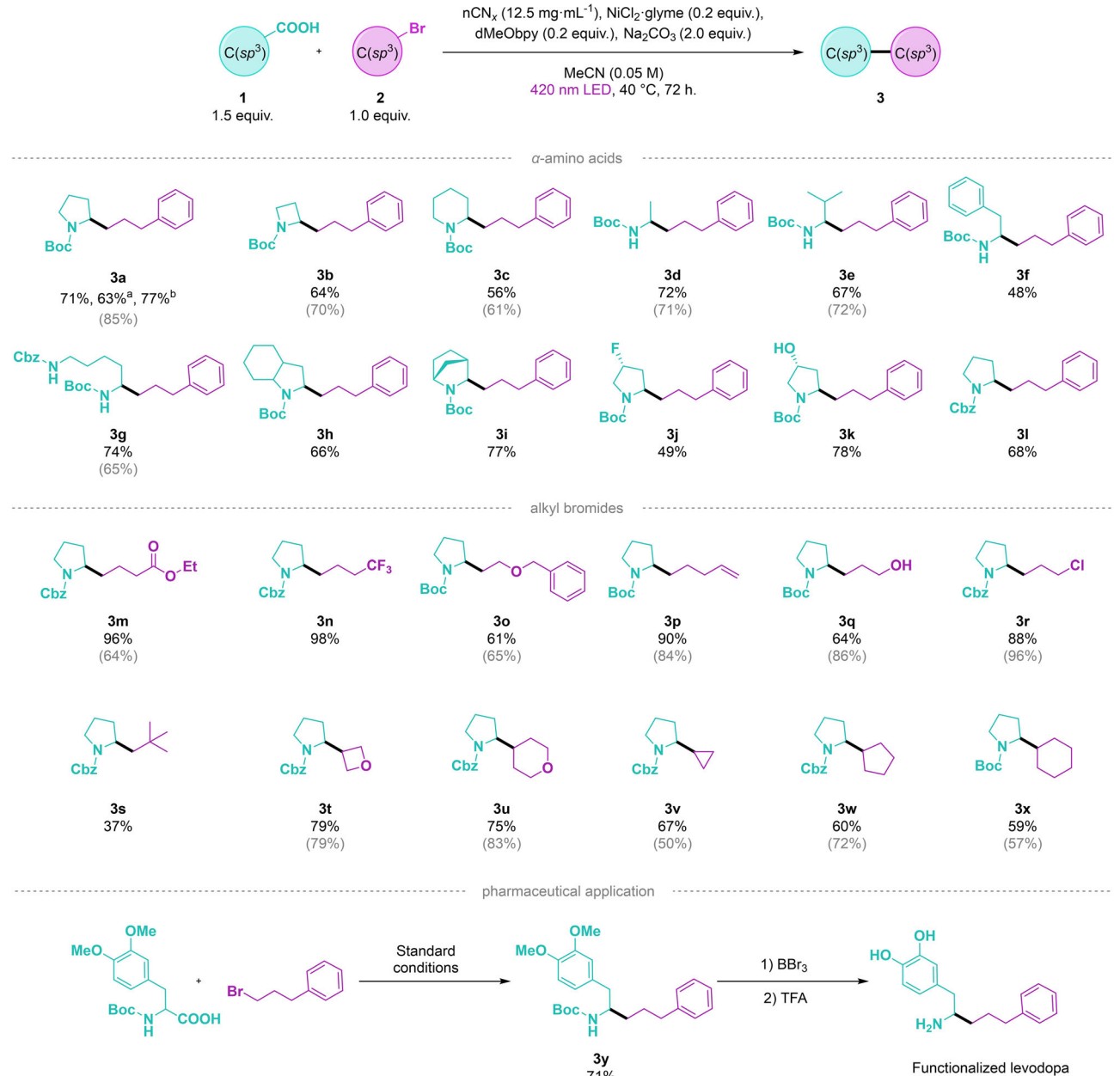

**Fig. 3 | Decarboxylative C($sp^3$)−C($sp^3$) cross-coupling reaction scope.** The characterization of the isolated products is described in the Supplementary Material. In gray between brackets, yields reported using homogeneous Ni complex and Ir photocatalyst[18]. [a]Reaction scaled up to 2 mmol by sizing up. [b]Numbering up strategy of 15 multi-batch reactions.

opportunities for innovation and differentiation in the pharmaceutical market[47]. The functionalized levodopa could be subsequently obtained through ether cleavage, followed by Boc deprotection.

Finally, the robustness and long-term stability of the $nCN_x$ catalyst were demonstrated through five recyclability tests, consistently obtaining the alkylated product **3a** with yields between 77% and 81% HPLC yield (Table S9). After each reaction, the $nCN_x$ catalyst was recovered by centrifugation, washed with ethyl acetate and water, and dried at 65 °C overnight. The recovery rate of the catalyst was quantitative in each instance. Post-catalysis characterization of the $nCN_x$ photocatalyst after five consecutive catalytic cycles was then conducted. BET adsorption isotherms demonstrated a surface area of 50 m²·g⁻¹ (Fig. S3a), and the observed increase in surface area was likely attributed to further exfoliation of the material caused by sonication during the washing process. XRD analysis of the same material

revealed two characteristic diffraction peaks at $2\theta = 13°$ and 27°, which were present also in the fresh material and corresponded, respectively, to the (100) and (002) planes of carbon nitride (Fig. S3b). Moreover, X-ray photoelectron spectroscopy (XPS) revealed no significant variations in the chemical environment of C and N before and after the catalytic tests, indicating the structural stability of the material under the applied reaction conditions (Fig. S3c). The C/N ratio was found to be 0.65, close to the ideal value of 0.7. Additionally, ICP analysis revealed that 0.2 wt.% of Ni adhered to the material after the five recycling tests (Table S10, entry 2), which is consistent with reported findings[37,48]. Nevertheless, the recycled photocatalyst was found to be inactive (Table S9, entry 6), and removal of the adhered Ni traces from $nCN_x$ *via* washing with 1 M HCl (Table S10, entry 3) did not impact its intrinsic photocatalytic performance (Table S9, entry 7). Scanning electron microscopy (SEM) revealed a reduction in particle size for the

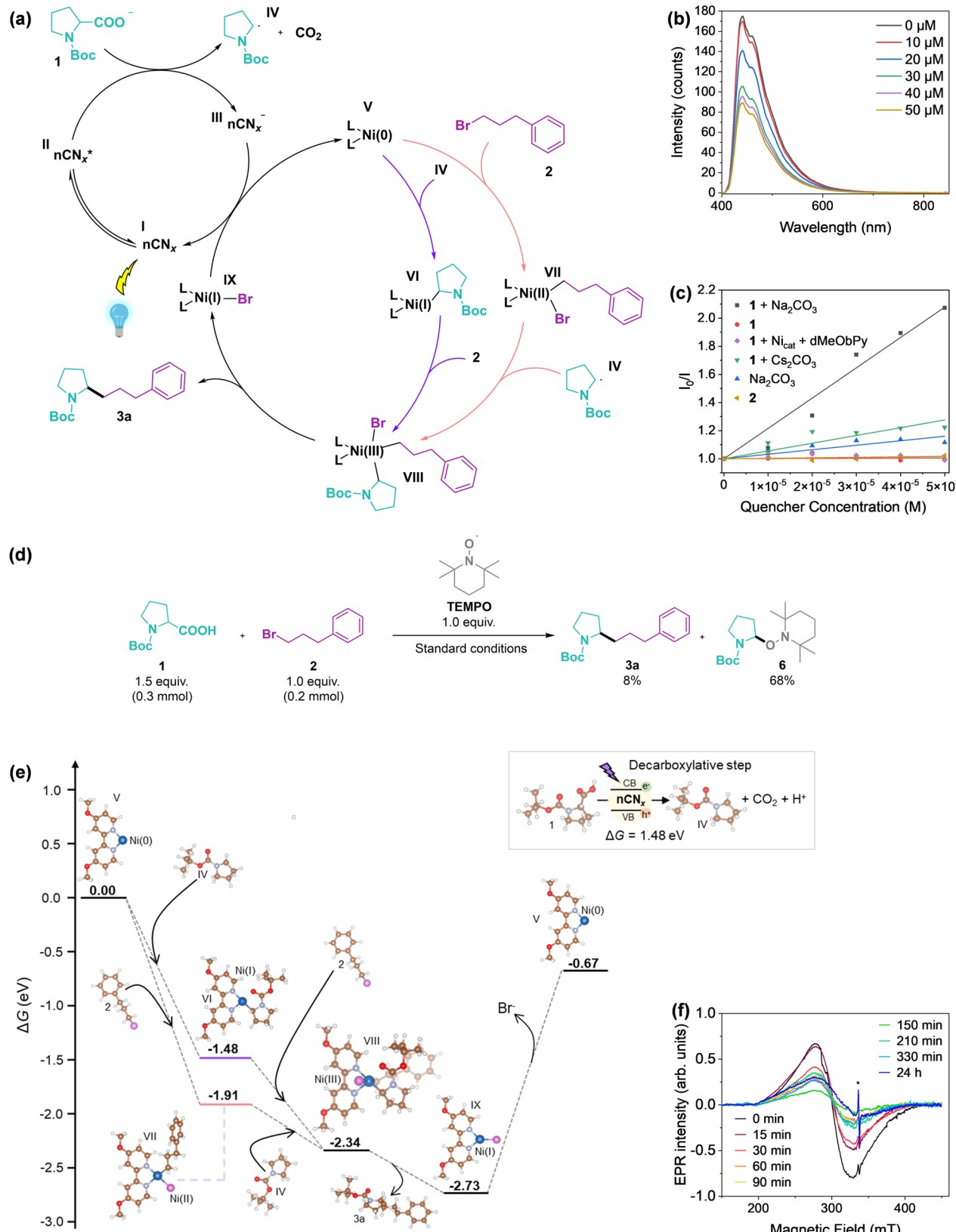

**Fig. 4 | Mechanistic investigations and energy profile of the dual catalytic system.** Proposed dual catalytic reaction mechanisms for the C–C cross-coupling (**a**), Stern-Volmer quenching of **1** with $Na_2CO_3$ (**b**), quenching plots of different species (**c**), radical trap experiment (**d**), DFT-calculated energy profile for the decarboxylative $C(sp^3)$–$C(sp^3)$ cross coupling reaction (**e**), and EPR evolution of the reaction (**f**). Brown, blue, light purple, pink, red, and white balls correspond to carbon, nickel, nitrogen, bromide, oxygen, and hydrogen atoms, respectively.

recycled material (Figs. S4 and S5), consistent with the increased surface area. Additionally, transmission electron microscopy (TEM) confirmed the retention of the nanosheet (lamellar) morphology after the catalytic cycles (Figs. S4 and S5).

After demonstrating the protocol's flexibility and the catalyst's stability, the mechanistic features of the dual catalytic cycle were investigated using a combined experimental and computational approach. A mechanistic cycle was initially hypothesized (Fig. 4a) in which the $nCN_x$ photocatalyst (intermediate **I**) is excited by light radiation to form intermediate **II**. This photoexcited photocatalyst could undergo oxidative SET to the deprotonated compound **1**, followed by dissociation of the neutral $CO_2$ moiety, resulting in the decarboxylated radical **1** (intermediate **IV**) and the reduced $nCN_x$ intermediate **III**. Based on this hypothesis, we have analyzed two possible pathways. In the first case, the Ni(0) catalyst (intermediate **V**) undergoes oxidative addition of the alkyl halide **2** to form the Ni(II) intermediate **VII**, which is followed by the oxidative trapping of intermediate **IV** to form the Ni(III) intermediate **VIII** (Fig. 4a, red pathway). This sequence follows the mechanism proposed by Molander and coworkers in the decarboxylative $C(sp^2)–C(sp^3)$[49]. Alternatively, the Ni(0) catalyst might first undergoes oxidative radical trapping to form the Ni(I) intermediate **VI**, followed by the oxidative addition of the alkyl halide **2** to obtain intermediate **VIII** (Fig. 4a, blue pathway). Reductive elimination of intermediate **VIII** results in product **3a** and Ni(I) intermediate **IX**, which then completes the catalytic cycle through reductive single-electron transfer with the reduced $nCN_x$ intermediate **II**.

To validate the mechanism, Stern-Volmer experiments were conducted to determine the different quenching constants, providing information about the interaction between photoexcited $nCN_x$ and various chemical species (Fig. 4c and Table S11). In fact, these experiments were crucial for understanding the reaction's selectivity with different bases. Our findings demonstrate a strong quenching interaction ($K_{sv} = 2.15 \times 10^4$) between $nCN_x$ and the $Na_2CO_3$-deprotonated **1**. In contrast, the quenching constant decreased significantly ($K_{sv} = 5.5 \times 10^3$) when $Cs_2CO_3$ was used (Fig. S6f). This difference may be translated to a more effective SET step with $Na_2CO_3$, which facilitates the decarboxylation process and ultimately favors the formation of the desired product **3a** over the byproduct **4a**, as the selectivity for the product formation is closely linked to the ability of the system to promote the decarboxylation step. Furthermore, the emission peak's shape and position remained unchanged upon the addition of the quencher, indicating that the local environment was unaffected by the presence of the quenching molecules (Fig. S6g).

It was then demonstrated that radical intermediate **IV** could be formed under our standard conditions (without **2**), as evidenced by the successful trapping of the hypothesized radical intermediate **IV** by the radical scavenger TEMPO, resulting in the formation of compound **6** (Fig. S7). Furthermore, when the standard conditions were reproduced in the presence of TEMPO, 8% of product **3a** was obtained. It was also observed that the radical intermediate **IV** was preferentially quenched by the radical scavenger, leading to the formation of 68% of compound **6** (Fig. 4d). These experiments provide evidence of the principle that $nCN_x$ could effectively decarboxylate **1** to form the radical, demonstrating that the radical intermediate plays a key role in the reaction mechanism.

Spin-polarized DFT calculations were performed to simulate the photodecarboxylation and subsequent C–C bond formation, and computational details are reported in the Supplementary Material. Specifically, transition states were not calculated due to errors when accounting for multiple reaction pathways and the solvation effects of leaving groups such as $Br^-$, which to date are challenging to capture thermodynamically. However, the thermodynamic effects were incorporated by considering the entropy correction at the reaction temperature of 40 °C. The reaction barriers were calculated using the thermodynamic approach developed by Nørskov and coworkers, which assumes the existence of reactions correlating the activations and free energies, and also accounts for the Brønsted–Evans–Polanyi relationship (BEP)[50–52]. Ni(0) was considered anchored to the 4,4′-dimethoxy-2,2′-bipyridine ligand. The Ni atom was bonded to two N atoms with a Ni–N bond distance equal to 1.87 Å (see Fig. S10 for a complete picture of the catalyst and its corresponding bonds and angles). The C–C coupling is an exergonic process with a total $\Delta G = -0.67$ eV. As discussed below, the chemical steps are energetically favorable except for the two involving the photoexcitation of $nCN_x$. In the first, the photoexcited hole promotes the decarboxylation of compound **1**. In the second, the photoexcited electron promotes the recovery of the catalyst through the reduction of nickel from Ni(I) to Ni(0) state.

The first process considered is the decarboxylation process of compound **1** (Fig. 4f). Based on the experimental results, it was hypothesized that sodium ions, due to their smaller size, may provide better stabilization of the transition state compared to other bases, thereby enhancing selectivity toward the formation of **3a**. To explore this, the intermediates formed through the interaction of **1** with various bases were modeled using DFT (Fig. S11). The analysis revealed a similar intermediate energy for $Na_2CO_3$ (1.24 eV) and $K_2CO_3$ (1.25 eV), both of which were more stable than in the absence of a base (1.48 eV). In contrast, a significant energy barrier was observed for DBU (2.31 eV). Notably, the sodium cation forms an intermediate that has a stronger and more localized basic environment compared to that of the potassium-based intermediate. This effect is primarily attributed to the smaller size of the sodium cation and its electronic interactions with the deprotonated intermediate **1**. Specifically, $Na^+$ leads to a higher selectivity towards product **3a** by stabilizing the intermediate through the formation of two bonds with oxygen atoms, each measuring ~2.21 Å in length (Fig. S12a). This tighter interaction favors the decarboxylation step and the selective formation of product **3a** over byproduct **4a**. Conversely, the $K^+$ interacts more weakly with the intermediate, forming three longer bonds with oxygen atoms: two at 2.59 Å and one at 2.70 Å (Fig. S12b). Due to these weaker and less stabilizing interactions, the reaction proceeds with reduced selectivity, resulting in the formation of both product **3a** and byproduct **4a** at comparable rates. The calculated band gap of $nCN_x$ using the PBE0 functional is 3.9 eV, which is slightly larger than the measured value. This discrepancy arises because accurately reproducing the band gap of semiconductors is a well-known challenge within DFT[28]. However, the PBE0 functional has been selected due to its ability to better account for electron correlation effects, making it a reliable method for our calculations despite the observed discrepancy[53,54]. The thermodynamic energy cost for the photoexcitation of $nCN_x$ is 1.48 eV and, and once photoexcited, the electrons in the conduction band (CB) of $nCN_x$ start to promote the decarboxylation process[55]. Two distinct pathways for the reaction were then modeled. In the first path, the catalyst initially reacts with the radical generated during the decarboxylation process, resulting in a Ni(I) state. This is followed by the adsorption of the alkyl halide reagent (**2**, R′–Br), during which the transition metal changes its oxidation state to a Ni(III). In the second path, the catalyst first reacts with the alkyl halide **2**, assuming a Ni(II) oxidation state, followed by the adsorption of the radical formed in the decarboxylation process, leading to a Ni(III) state. Both pathways lead to an intermediate where Ni(III) coordinates both R, Br, and R′. The analysis of the free energy profiles suggests that the formation of Ni(II) adduct **VII** is more thermodynamically favorable, with a $\Delta G = -1.91$ eV compared to a –1.48 eV for the alternative pathway. This is illustrated in the energetic profile shown in Fig. 4e. Details of the mechanism for the alternative pathway are reported in the Supplementary Material (see Figs. S13 and S14). For the thermodynamically-favorable step, after adsorption of R′–Br, the metal is four-coordinated in a nearly

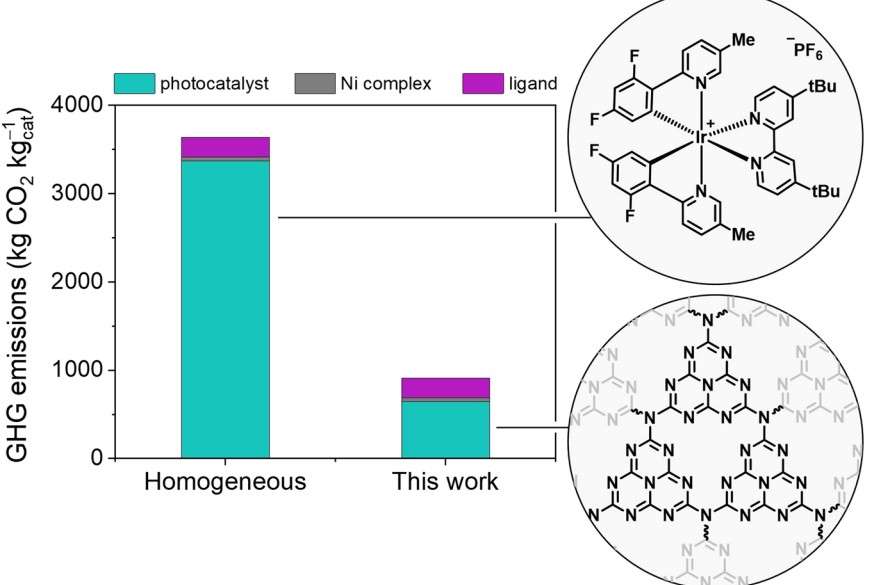

**Fig. 5 | Comparative life cycle impact of homogeneous and semi-heterogeneous protocols.** Global warming impact (kg $CO_2$-equiv./kg catalyst component) for the manufacturing of the two photocatalytic technologies discussed in this study. The homogeneous catalytic system is based on the use of Ir[dF($CF_3$)ppy]$_2$(dtbbpy)PF$_6$[11], while the semi-heterogeneous system developed in this paper exploits nCN$_x$. Ni complex = NiCl$_2$·glyme, ligand = dMeObpy.

square planar configuration, according to its Ni(II) character. The two Ni–N bond distances are slightly elongated with respect to the bare catalyst (1.87 Å), 1.90 Å, and 1.99 Å, respectively. There is a direct Ni–Br interaction with a distance equal to 2.32 Å, and a Ni–C bond at 1.97 Å (Fig. S10c).

The next step is the adsorption of the radical, as shown in Fig. 4e (intermediate **VIII**). This step is also exergonic (−0.43 eV), during which Ni changes its coordination. Here, Ni is five-coordinated, with a trigonal pyramid shape. The two Ni–N bonds are further elongated to 1.99 Å and 2.07 Å, as well as the Ni–Br bond distance goes to 2.52 Å. There are two Ni–C bonds with distances of 1.97 Å and 1.99 Å (Fig. S10d). Specifically, the Ni–Br bond distance is elongated by 9% with respect to intermediate **VII**, while the Ni–C distance corresponding to the R′–Br group remains almost the same (1.99 vs. 1.97 Å) (Fig. S10c, d). The formation of the third and final intermediate involves the C($sp^3$)–C($sp^3$) coupling process between the carboxylic acid and alkyl halide group, leading to the release of the product along with the catalyst and Br−. The process is again exergonic, with $\Delta G = -0.39$ eV. Ni(III) is reduced to Ni(I), and the Ni–Br bond distance is much shorter at 2.23 Å (Fig. S10e). Finally, the last step involves the interaction of intermediate **IX** with a hole in the valence band of the reduced nCN$_x$, leading to the release of the Ni catalyst complex and Br−. To understand the role of the Boc-protecting group, the reaction profile was recalculated in the absence of the Boc group (Fig. S15). The results revealed that the reaction energy profile is equivalent with (Fig. 4e) and without Boc, although more stable intermediates are formed when comparing Boc-protected and non-protected proline. As a result, the overall reaction mechanism remains energetically equivalent in both cases, and our mechanistic conclusions are unchanged, with the Ni(II) intermediate **VII** still being more stable than Ni(I) intermediate **VI**, as shown in Figs. 4e and S15. This outcome is expected since the Boc acts as a protective group in proline, preventing unwanted reactions of its amine functionality and influencing the local environment of the reactant and intermediate states. It should be mentioned that the lack of significant changes in the reaction profile observed in the DFT calculations could be attributed to the nature of DFT as a local optimizing method. While DFT captures the electronic structure, it may not fully account for long-range effects or

subtle changes from the conformational flexibility or dynamic interactions of the Boc group in a real reaction setting. Details of the intermediates and mechanisms for the alternative pathways where the Boc is not considered are reported in the Supplementary Material (Figs. S16–18). EPR spectroscopy was further employed to monitor the oxidation state of Ni ions during the reaction. Aliquots of the reaction mixture were sampled at different time intervals and analyzed. An EPR spectrum centered at $g = 2.24$ was observed, consistent with a high-spin ($S = 1$) Ni(II) species (Fig. S9). The spectral intensity decreases over time, reaching a steady state after ~2 h (Fig. 4f). No evidence of Ni(I) species ($S = 1/2$), characterized by a distinct EPR spectrum, was observed. These results corroborated that under reaction conditions, high-spin Ni(II) species were involved in the photocatalytic cycle[27], in line with the DFT calculations.

The work demonstrates the potential for replacing Ir-based organometallic complexes with carbon nitride nanosheets. With the growing interest in sustainability metrics, we finally investigated the GHG emissions resulting from the manufacturing of the catalytic systems (Fig. 5). In the semi-heterogeneous case developed in this work, the Ir photocatalyst is essentially substituted with carbon nitride, a metal-free, polymeric photocatalytic material synthesized from melamine, an abundant and inexpensive starting reagent. The Ni complex and corresponding dMeObpy ligands are used in the homogeneous and semi-heterogeneous systems and are required for the overall catalytic cycle.

The manufacturing of the initial Ir catalysts for the dual photocatalytic process produced 3370 kg $CO_2$-equiv./kg of Ir[dF($CF_3$) ppy]$_2$(dtbbpy)PF$_6$, while the modification of the photocatalyst component alone accounts for a drop in GHG emissions of just over 80% (645 kg $CO_2$-equiv./kg of nanosheet-based carbon nitride material, nCN$_x$). The main driver for the decrease in GHG emissions is the avoidance of the Ir element in the heterogeneous photocatalyst, as the energy input required for the extraction of Ir is the key contributor to total GHG emissions. The calculation thus quantitatively supports the sustainability of the method developed in this paper.

In summary, we have successfully developed a highly effective approach for C($sp^3$)–C($sp^3$) cross-coupling reactions using a photocatalytic system based on carbon nitride (nCN$_x$) and a nickel complex.

This innovative method, driven by visible light and utilizing only earth-abundant elements, achieves a remarkable maximum yield of 77% for the target compound **3a**. Our process is both selective and efficient, eliminating the need for preactivation of the photocatalyst, a significant advantage that simplifies the reaction setup and enhances practicality. The reaction conditions were optimized to enable broad applicability across a wide range of substrates, including cyclic and linear $\alpha$-amino carboxylic acids as well as various alkyl bromides. This versatility facilitated applications such as the functionalization of pharmaceuticals like levodopa or lysine. Furthermore, mechanistic studies incorporating radical trapping, Stern-Volmer analysis, EPR spectroscopy, and DFT calculations offered valuable insights into the catalytic cycles involving $nCN_x$ and nickel catalysts. Overall, the study underscores the potential of using carbon nitride and nickel complexes as sustainable and green alternatives to traditional iridium-based photocatalysts. The simplicity, efficiency, and reliance on non-precious materials make this approach a significant advancement in developing environmentally friendly synthetic methods for pharmaceuticals and other fine chemicals.

## Methods

### Material characterization

The $CN_x$ material was prepared through the calcination of melamine. The material was characterized via $N_2$ physisorption on a 3 P Sync 400 instrument at $-196\,°C$ to evaluate the porous nature and surface areas of the samples. An outgassing step was carried out at $150\,°C$ for 24 h before the textural measurements to remove any residual moisture or adsorbed contaminant from the catalytic surfaces. The specific surface area was determined through the Brunauer-Emmett-Teller (BET) method. The porosity and pore distribution were calculated by using the model of quenched solid density functional theory (QSDFT) for $N_2$ adsorbed on carbon at 77 K. The material phase purity and crystallinity were evaluated through X-ray diffraction (XRD) using a Bruker D2 Phaser X-ray diffractometer equipped with a Cu K$\alpha$ radiation source ($\lambda = 0.15405\,nm$) within the $5-60°\ 2\theta$ range. The samples, in powder form, were placed on a flat holder and directly analyzed in air without any further treatment. The XRD diffractograms were acquired with a $2\theta$ step size of $0.016°$ and a counting time of 0.4 s per step. CHN analysis was performed by combustion on a Vario Micro Elemental Analyzer, following their combustion at high temperatures ($1000\,°C$). ICP-OES analysis was performed using a PerkinElmer Optima 8300 equipped with a photomultiplier tube detector. After dissolving the catalyst in a strong acidic medium and nebulizing it into a fine aerosol, an inductively coupled plasma torch was used to generate excited atoms and ions. UV-Vis spectra were recorded in ambient conditions using a UV-Vis-NIR Varian Cary 5000 spectrophotometer equipped with a reflectance sphere. The sample was introduced into an appropriate cell for solid materials. Barium sulfate ($BaSO_4$) was used as a reference material for the DRS measurements. The collected reflectance value was successively converted into pseudo-absorbance ($A$) through to the relation $A = -\log(R)$, where $R$ denotes the reflectance values collected in the DRS measurements. XPS measurements were carried out with an X-ray gun Mg K$\alpha$ radiation (1254.6 eV) using the CISSY end-station under ultra-high vacuum (UHV) at $1.5 \times 10^{-8}\,Pa$, equipped with a SPECS XR 50 and Combined Lens Analyzer Module. The binding energy scale and Fermi level were calibrated using a gold film. The ultraviolet photoelectron spectroscopy (UPS) spectra were acquired with a He I (21.2 eV) radiation source. The detector was a combined lens with an analyzer module, thermoVG (TLAM). TEM bright-field images were acquired using a Philips CM200 electron microscope operating at 200 kV equipped with a Field Emission Gun filament. A Gatan US 1000 CCD camera was used, and $2048 \times 2048$ pixels images with 256 gray levels were recorded. Morphological analyses were performed by SEM, using an SEM Zeiss EVO 50 EP.

### Reaction set-up

Catalytic reactions were conducted in an 8 mL vial, where the carboxylic acid **1** (0.3 mmol, 1.5 equiv.), $NiCl_2$·glyme (0.04 mmol, 0.2 equiv.), dMeObpy (0.04 mmol, 0.2 equiv.), $nCN_x$ (50 mg), and $Na_2CO_3$ (0.4 mmol, 2.0 equiv.). The mixture was dissolved in MeCN (4 mL) and, to the suspension, the corresponding alkyl bromide **2** (0.2 mmol, 1.0 equiv.) was added. The vial was closed with a screw cap rubber septum and inserted in the photoreactor. The suspension was vigorously stirred under purple light (420 nm) at $40\,°C$ for 72 h. After this time, the crude was filtered off through a PTFE syringe filter (0.45 μm) and washed with a saturated aqueous solution of $NaHCO_3$. The aqueous layer was washed with more EtOAc (3 × 5 mL). The organic layers were collected and dried over $MgSO_4$. The organic phase was thus purified via silica gel column chromatography to give product **3**. Additional details about compound purity and sourcing, as well as individual syntheses for the substrate scope are given in the Supplementary Materials.

### Mechanistic investigations

Stern-Volmer experiments were conducted in a 2 mL suspension of $nCN_x$ in acetonitrile (0.25 mg mL$^{-1}$) using a 1 cm quartz cuvette with stirring. A 375 nm CW laser (550 μW) excited the sample, and emitted light was filtered through a dichroic mirror and detected by a spectrometer. A 1000 μM stock solution of the quencher molecule in water-acetonitrile was incrementally added (10 μM steps). EPR spectroscopy was also used to study the oxidation state of Ni ions. Ni(II) ions ($S = 1$) in tetrahedral coordination were detected via a broad signal at $g = 2.245$ in the EPR spectrum at 77 K. The reaction mixture's initial spectrum matched the $NiCl_2$·glyme precursor, indicating the presence of EPR-active Ni(II) species. Spin-polarization DFT calculations were performed using VASP. Core electrons were treated with the Projector Augmented Wave pseudopotential method. The PBE exchange-correlation functional and Grimme's DFT + D3 scheme with Becke-Johnson damping accounted for van der Waals interactions. The GHG emissions of the catalytic systems were assessed using retrosynthetic analysis to deconstruct the respective components into more fundamental building blocks and associated energy requirements, considering molar ratios, heat requirements, and reaction yields for each step involved in the synthesis of the catalytic units. The contribution of each individual component to the GHG emissions was retrieved from the Ecoinvent 3.9.1 cut-off and Carbon Minds databases. Further details on the Methods are given in the Supplementary Material.

## Data availability

The Supplementary Information and Supplementary DFT data are available and free of charge. Source data are provided with this paper.

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

## Acknowledgements

MMdVI gratefully acknowledges the European Commission for the Marie Skłodowska-Curie doctoral fellowship (101073089, GreenDigiPharma). G.V. thanks the funding from the European Commission's Horizon Europe research and innovation program (101057430, SusPharma). The postdoctoral position of L.A.C. was funded through a European Research Council grant (101075832, SAC_2.0). G.D.L. received funding through the PRIN project "UNDERSAC" (project code 2022LRPSTS) by the Italian Ministry for Universities and Research (MUR). A.P. and A.O. acknowledge that this work is part of the "Technologies for Sustainability" flagship program of the Istituto Italiano di Tecnologia. The authors extend their acknowledgements to Nicolò Allasia for support in characterizing the properties of carbon nitride, and Dr. Mark A. Bajada for discussions. We also acknowledge CINECA for providing high-performance computing resources and support under the ISCRA initiative.

## Author contributions

G.V. conceived and coordinated the work. M.M.d.V.I. prepared the carbon nitride catalyst, characterized its properties, and conducted catalytic reactions, Stern-Volmer experiments, and EPR mechanistic studies. V.L. and M.C. supported with EPR mechanistic studies, A.O. and A.P. supported with Stern-Volmer experiments, and L.A.C. and G.D.L. performed DFT calculations. All authors contributed to writing this manuscript and approved its final version.

## Competing interests

The authors declare no competing interests.
