## [Peer Review File · Nature Communications]

Photocatalytic C(sp³)-C(sp³) cross-coupling of carboxylic acids and alkyl halides using a nickel complex and carbon nitride

Corresponding Author: Professor Gianvito Vilé

Version 0:

Reviewer comments:

Reviewer #1

(Remarks to the Author)

I evaluated the manuscript by Gianvito Vilé and co. entitled "Photocatalytic C(sp³)-C(sp³) cross-coupling of carboxylic acids and alkyl halides using a nickel complex and carbon nitride" with great interest. In the report, the author developed an approach merging nickel catalysis and heterogeneous photoredox catalysis, through the use of carbon nitride nanosheets (nC₃N₄), to form a C(sp³)-C(sp³). The reaction proceeds via the use of carboxylic acid containing compounds (mostly cyclic 4, 5 and 6-membered rings) that undergo decarboxylation to form the corresponding alkyl radical that undergoes reaction with a halogenated substrate. Overall, the study is nicely executed but unfortunately, as stated by the authors, and as analyzed by this reviewer, the main advancement of the present study is the substitution of iridium photosensitizers for more abundant carbon nitride photosensitizers. Whereas this is a noble goal, since the reaction on some of these derivatives are already known and reported using Ir(III) photosensitizer, this study only focuses on repeating this transformation using another abundant photosensitizer. It is this reviewer's opinion that such iterative study should not warrant publication in Nature Communications as they solely rely on changing the photosensitizer. To emphasize this point, the mechanism as proposed relies on the oxidation of the proline derivative. As such, other organic photosensitizers such as acridinium and pyrylium derivatives could also be as efficient for this transformation and it is not conceivable to publish in Nature Communication each time a new photosensitizer is screened, unless additional mechanistic information are obtained, which is not the case here. I would therefore recommend publication in a more specialized journal such as Org. Lett. for example.

Here are also specific points to consider, in no particular order:

- Figure 1: "High temperature" should be "High temperature"
- K-PHI is not defined in the text.
- The authors mention purple LED at 420 nm and then Blue LED at 420 nm (on page 7 and 8). This should be corrected.
- Table 1: The structure of the proline derivative is wrong (it should be COOH and not CH₂OOH)
- The authors should emphasize also or compare the yields with those reported by MacMillan in 2016, which for 25% of them are roughly identical.
- What is the recovery and recyclability of C₃N₄? The authors have a short paragraph on it that is immediately re-engaged in the same reaction but I wonder how much would be recovered at the end of the reaction and if it would operate as efficiently over 5 new reactions.
- I could not judge the DFT calculations as it is beyond my area of expertise but I would recommend that the author carry out some excited-state quenching experiments to corroborate their mechanism (although it is identical to the one proposed by MacMillan). Stern-Volmer experiments using carbon nitride and the different partners (proline, Nickel, Proline and nickel together) should be investigated to investigate possible quenching routes.

Reviewer #2

(Remarks to the Author)

sp³ C-sp³ C cross-coupling of carboxylic acids and alkyl halides presents a strong tool for the synthesis of a diverse array of functionalized drug molecules. This work modified the reported Ni-Ir process by replacing Ni with PCN, and the results are important and interesting in some degree. However, the following issues would be seriously addressed before the possible

acceptance: 1) the roles of CN_x, Ni, ligand, base can be further explored. 2) The yield of 3a increases, and that of 4a decreases as the Na₂CO₃ increases from 1 to 2 equiv; while the 3a yield decreases but the 4a yield increases as the Na₂CO₃ increases from 2 to 3; and then 3a increases but 4a decreases as the alkaline increases from 3 to 4. Please explain. 3) Compare Table 8 and Table 1, there are too many same control results are repeatedly presented in the two tables. 4) Many related reports regarding of sp³ C- sp³ C cross coupling are missed, ie. ACS Med. Chem. Lett. 2018, 9, 7, 773–777. 5) The description regarding of Figure 1 is different from the Figure. Figure 1a is not for a photocatalytic homogeneous decarboxylative C(sp²)-C(sp³) coupling using Ni catalyst and Ir photocatalyst; that for Figure 1b is also wrong. 6) Where is the result for recyclability? 7) some experimntal proof for the change of Ni chemical state is better to be presented? 8) Please explain how to obtain the compared CO₂ emission in Figure 5.

Reviewer #3

(Remarks to the Author)

This manuscript describes dual Ni/carbon nitride catalyzed C(sp³)-C(sp³) coupling of carboxylic acids and alkyl halides. Although replacing noble Ir by carbon nitride has been demonstrated previously, this study offers a more practical protocol for dicarboxylic C(sp³)-C(sp³) coupling, in terms of cost, catalyst recycle and low CO₂ emission.

(1) In an earlier publication, the authors reported carbon-oxygen coupling from the same coupling partners employing single atom Ni anchored on carbon nitride. Carbon-oxygen coupling also proceed with different yield, depending on conditions. It is worth to find the underlying reasons for different coupling manners, which will increase the scientific impact and guide catalyst development.

(2) The statement “molecular insights into the role of nickel single atoms in facilitating photodecarboxylation and subsequent C–C bond formation”, where single atom is used for Ni in the bpy coordinated complex might not be suitable.

(3) The manuscript requires substantial reorganization. The characterizations and condition optimizations should be condensed, and many details are better suited for inclusion in the Supplementary Materials

(4) The photoreaction conditions should be provided in detail (solution volume, incident light intensity at different wavelengths); MeCN (x M) is confusing, whose concentration in the bracket indicate? MeCN or alkyl bromide?

(5) Please provide the data for catalytic performance of the recovered catalyst, together with the structural characterizations (XRD, TEM, and the possible Ni aggregates)

(6) Ni(0) is proposed to interact with alkyl halide or Intermediate IV. The initial state of Ni is Ni(II). And there are controversies on the active forms of Ni (Ni(I) or Ni(0)). The authors may want to track the evolution of Ni on CN experimentally.

(7) Beside free energy changes, the transition states and the corresponding barriers in the two proposed pathways should be provided to support the assignment of the dominant one.

(8) Calculation and references for GHG emissions should be provided.

Reviewer #4

(Remarks to the Author)

In this manuscript, Gianvito Vilé et al. present a study on C(sp³)-C(sp³) cross-coupling in a semi-heterogeneous system using an nCN_x photocatalyst. While the exploration of such catalysts and their application in cross-coupling reactions is undoubtedly significant, and the concept of utilizing nCN_x as a heterogeneous photocatalyst is intriguing, the manuscript falls short in distinguishing this approach from prior research. For instance, in a previous study (Nat. Synth. 2023, 2, 1092-1103), Gianvito Vilé already employed a Ni single-atom catalyst for C-O coupling but did not adequately elucidate the rationale behind the occurrence of decarboxylation in this study. A more in-depth investigation into the decarboxylation mechanism is needed, particularly in light of earlier reports (Angew. Chem. Int. Ed. 2024, 63, e202405902; Adv. Synth. Catal. 2020, 362, 3898-3904). Additionally, it is well-established that homogeneous nickel catalysts are prone to deactivation in dual photoredox/nickel-catalyzed processes, as highlighted in a previous study (Nat. Catal. 2020, 3, 611-620). This manuscript does not sufficiently address this critical issue. I do not recommend its publication in Nature Communications. The manuscript should include more comprehensive studies, incorporating extensive experimental work and computational analysis, to elucidate a mechanism and demonstrate the clear advantages of semi-heterogeneous decarboxylative cross-coupling.

Comment #1: The previous studies referenced on page 2 do not correspond accurately with the information presented in Figure 1. For instance, the study shown in Figure 1c is from 2016, not 2014 as stated in the text. Please ensure that all studies mentioned in the text are correctly aligned with those depicted in the figure, and address any typographical errors.

Comment #2: The authors optimized several photocatalysts, including nCN_x, gCN_x, mpg-CN_x, K-PHI, and recovered nCN. However, the manuscript lacks sufficient characterization data to substantiate the comparison and validation of nCN_x against the other photocatalysts. Although nCN_x was prepared through a thermal exfoliation process from gCN_x, no data are provided to describe the morphology, thickness, or surface area of nCN_x relative to gCN_x. Therefore, it is essential to include comprehensive characterization data for all the photocatalysts in Figure 2, such as BET, XRD, AFM, and UV analyses. Additionally, TEM, SEM, XPS, and FT-IR data should be provided for each photocatalyst. Furthermore, the method used to calculate the reported C/N ratios (‘0.61-0.67’) should be clearly explained.

Comment #3: In this study, the authors present an optimization of a photocatalyst, identifying nCN_x as the optimal choice. However, several concerns arise regarding the rationale behind selecting nCN_x over other catalysts. The authors suggest that mpg-CN_x is less effective due to its higher solution viscosity, which they claim reduces the availability of active sites. This assertion, however, lacks sufficient experimental evidence or comprehensive characterization of the photocatalysts in question. While it is acknowledged that mpg-CN_x may exhibit increased viscosity, this factor alone does not provide a strong

basis for dismissing its catalytic potential. In fact, mpg-CN_x typically has a larger surface area, which is generally associated with an increased number of active sites, as documented in the literature (Chem. Eur. J. 2015, 21, 526–530; J. Phys. Chem. C 2012, 116, 19644–19652; Angew. Chem. Int. Ed. 2015, 54, 12868–12884). This contradicts the authors' claim that mpg-CN_x is less suitable due to fewer active sites. The manuscript lacks detailed characterization data, such as BET surface measurements, pore size distribution, or SEM/TEM imaging, which are essential for substantiating the claim that mpg-CN_x has a reduced number of active sites or is significantly impacted by viscosity-related issues. Moreover, the authors report comparable reaction yields between CN_x (3a 41% and 4a 23%) and mpg-CN_x (3a 34% and 4a 18%), indicating that mpg-CN_x could be a viable alternative. The rejection of its efficacy based on unproven assumptions about solution viscosity and active site availability is unconvincing.

Comment #4: In addition to the concerns mentioned above, there is an inconsistency in the manuscript regarding the mention of "boron-doped mpg-CN_x." This material is introduced without prior context or discussion in the manuscript, leading to confusion. The authors should ensure that all materials discussed are properly introduced and contextualized.

Comment #5: The manuscript includes product yield data that have been confirmed using HPLC calibration. However, the authors have not provided the detailed evidence of the HPLC calibration curves that were used. To ensure scientific reproducibility, it is imperative that the authors provide comprehensive details regarding the calibration process. Furthermore, there are notable inconsistencies between the product yield values reported in the main text and those presented in the Supporting Information (SI). These inconsistencies call into question the credibility of the data and must be addressed. The authors should conduct a thorough review and make the necessary corrections to ensure consistency across the manuscript. Additionally, the SI lacks NMR spectra for several compounds discussed, despite the fact that these are typically required to confirm the synthesized products. The authors must provide the NMR data for all relevant compounds to support the claims made in the manuscript.

Comment #6: This paper presents DFT calculation data to explain the Ni-catalyzed reaction pathway. However, the DFT data presented, which uses proline, does not offer a sufficiently reliable explanation for the mechanistic pathway. Based on the substrate scope study in this paper, the reactivity and selectivity are influenced by the substituents, with even methyl-substituted proline showing no reactivity. Therefore, it would be more appropriate to present DFT calculation results using Boc-protected proline rather than proline. Additionally, the Ni-catalytic cycle has already been extensively addressed in previous studies (Nature 2016, 536, 322–325; J. Org. Chem. 2024, 89, 11136–11147; Science 2014, 345, 437–440). It would be beneficial to develop data that provides deeper mechanistic insights into decarboxylation beyond esterification.

Comment #7: The authors present GHG emission data derived from the equations described in the 'Energy Calculations' section of the Supplementary Information (SI) to support the sustainability of carbon nitride compared to Ir-based complexes. However, the mentioned 'Energy Calculation' section is absent from the SI. To substantiate the sustainability advantages of carbon nitride, it is recommended that the authors provide a detailed calculation process along with reliable references.

Reviewer #5

(Remarks to the Author)

The authors described the use of graphitic carbon nitride as a photocatalytic system to drive cross-coupling between alkyl halides and carboxylic acids. Mechanistic studies are also reported. In my view, the manuscript has been well written and the results are sound. Thus, I recommend its publication after having addressed the points below.

1. Additional references regarding the use of carbon nitride-based photocatalytic systems for the functionalization of organic compounds should be included in the introductory section, for instance: Science 365, 360–366 (2019), Angew. Chem. Int. Ed., 2023, e202313540, ACS Catal. 2023, 13, 13414–13422, Sci. Adv., 2020, 6, eabc9923, ACS Catal. 2024, 14, 11308–11317, ACS Nano, 2021, 15, 3621–3630, Chem. Sci., 2022, 13, 9927, Nature Catalysis, 3, 611–620 (2020), Adv. Sci. 2023, 10, 2303781, among others.
2. The reaction scope should be expanded. Is it possible to use fatty acids, alkyl chlorides and alkyl triflates as starting materials? Moreover, the generality of the photocatalytic system with respect to more synthetically useful organic substrates, namely natural products or active drugs, should be addressed. For instance, is it possible to use other natural products or bio-active molecules containing carboxylic moieties as substrates?
3. Is it possible to perform the model reaction in a gram scale?
4. I suggest to include a general procedure for the photocatalytic experiments along with a picture of the reaction set-up and its description within the SI.
5. The authors should better characterize the photocatalyst after a series of catalytic cycles.
6. I suggest an improvement on the structure of the manuscript due to the presence of typos (e.g., structure compound 1 within table 1)

Version 1:

Reviewer comments:

Reviewer #1

(Remarks to the Author)

I commend the authors for the quality and seriousness of their revisions. The manuscript is much improved and I felt like the

author responded professionally to all the comments. I have a very few minor comments:

In the novel figure 4, panel B, the "micro" symbol does not display properly, and only "u" is readable.

In Figure 4, panel C, the authors mislabeled the Y-axis, and I should be I₀/I and not I/I₀. In addition, compound 2, 1 and 1+ Nicat + dMeObpy (probably mislabeled as well), seem to have a negative slope that the author did not comment on. A negative slope would imply that the photoluminescence intensity increases as the quencher is added.

Still in figure 4, I would recommend forcing the intercept of all lines at (0;1), as this is a "true" datapoint according to the Stern-Volmer equation, $I_0/I = 1 + K_{sv}[Q]$, so when $[Q] = 0$, I_0/I should be 1.

Reviewer #2

(Remarks to the Author)

I read through the revised draft and the response to comments carefully, and find that the issues were basically addressed, and now can be acceptable.

Reviewer #3

(Remarks to the Author)

In the revised manuscript, the authors provided more insights into the C-C coupling (other than the previously reported C-O) by systematically excited state quenching, photon energy/base/solvent dependence and calculation. Other issues are also addressed.

Reviewer #4

(Remarks to the Author)

In the revised manuscript, Gianvito Vilé and colleagues have addressed some of the issues raised in previous comments. While the updated version demonstrates improvements, several critical concerns remain, rendering the manuscript unsuitable for publication in Nature Communications. The manuscript still lacks a thorough characterization of the materials used, which is essential for validating the reported material properties. Moreover, the choice of a semi-heterogeneous system requires a stronger justification to establish the significance of the findings. Additional concerns stem from experimental aspects, particularly the "numbering-up" scale-up process, which raises doubts about the reliability and scalability of the results. Consequently, despite the authors' notable efforts, I regret to recommend against publication in Nature Communications. Detailed comments are provided below:

Comment #1: The manuscript does not sufficiently highlight the advantages of the semi-heterogeneous system through a detailed comparison with other catalytic systems, such as homogeneous and single-atom heterogeneous systems. While the authors mention that the semi-heterogeneous system mitigates catalyst deactivation via stable interactions between the photocatalyst and nickel species, the manuscript lacks a clear and rational explanation to support this claim.

Comment #2: Although the authors stated that they included additional characterizations, critical analyses such as TEM, SEM, XPS, and AFM remain missing. The explanation of catalyst screening results is also insufficient to justify the selection of nCNx as the optimal material. Characterization data for the recovered nCNx are particularly lacking. For instance, TEM images and EDS mapping are necessary to determine whether Ni aggregates have formed. Despite claims that such data were included, they are absent from both the manuscript and the supplementary information.

Comment #3: For gram-scale synthesis, the authors employed a numbering-up strategy instead of scaling up the system size. This approach is unconvincing for demonstrating the scalability of heterogeneous semiconductors. Moreover, the numbering-up experiments were conducted on an even smaller scale (0.2 mmol) than the optimized conditions (0.3 mmol). As a result, the manuscript does not sufficiently support the claim that nCNx is a sustainable, efficient, and cost-effective alternative to traditional iridium-based photocatalysts, especially from an industrial perspective.

Comment #4: The authors propose that energy differences resulting from interactions between 1 and the base explain variations in product yields. This rationale is plausible in cases without a base (1.48 eV) or with DBU (2.31 eV). However, the manuscript fails to address the significant discrepancies between Na₂CO₃ and K₂CO₃ results. Despite their similar intermediate energies (1.24 eV for Na₂CO₃ and 1.25 eV for K₂CO₃), their yields differ substantially, with Na₂CO₃ yielding 33% of 3a and 17% of 4a, while K₂CO₃ yields 34% and 44%, respectively. Additional justification or experimental data is needed to clarify these differences.

Comment #5: Previously, it was suggested to provide detailed descriptions of the "Energy Calculations" for GHG emission data. However, the revised manuscript still lacks a clear explanation of the calculation procedures and data references. To emphasize the environmental advantages of the semi-heterogeneous system over homogeneous systems, a detailed explanation of how the GHG emission data were derived is essential.

Comment #6: Several inconsistencies in the manuscript undermine its reliability. These include: (1) Comparison of substrate yield with a prior study (MacMillan, 2016), despite differing experimental conditions (Ir system: room temperature, 48 h), which makes the comparison invalid. (2) Unexplained changes in nCNx surface area from 12 to 23 m²/g, raising questions about synthesis reproducibility. (3) Conflicting details about the washing procedure for the recovered catalyst. The supporting information states the catalyst was washed with ethyl acetate and water, while the manuscript and prior responses mention acetonitrile.

Comment #7: Several minor typos and formatting errors persist in the manuscript. For instance, the author's name is misspelled as "MCMillan" instead of "MacMillan" on page 3. Additionally, discrepancies between figure and data references within the manuscript need to be corrected.

Reviewer #5

(Remarks to the Author)

The revised version of the Manuscript has been well organized and the results are very interesting. I personally found this version of the manuscript more detailed and clear. In addition, the authors addressed most of the reviewers' comments satisfactory. Thus, I think that the new version of the manuscript is now suitable for publication in Nature Communications.

Version 2:

Reviewer comments:

Reviewer #1

(Remarks to the Author)

I was personally satisfied by the changes carried out last time by the authors. I have however to stress that I agree with reviewer 4, that a "numbering-up" approach is, in my opinion, not a proper scale up, but rather represents a reproducibility experiment from which average yields and standard errors can be obtained. I understand the rationale from the authors about light penetration, Beer-Lambert law etc, but a true scale up also implies accounting for these issues and showing applicability and practicability for industrial developments. Light penetration is often industrially tackled by multiplying irradiation sources, increasing fluence, inserting irradiation systems (light tubes) inside the reaction reactor etc. I am not sure if a scale up is desperately needed, but I feel like the author should acknowledge the limitation of the "numbering-up" approach that they propose.

Reviewer #4

(Remarks to the Author)

Some of the previous comments have been addressed, and the revised manuscript shows improvement. However, several important issues remain insufficiently explained. I recommend the authors provide further clarification based on the following comments. I will reconsider the publication of the manuscript once these concerns are fully addressed.

Comment 1:

It remains unclear whether byproduct 4a was formed when the $\text{Ni}_1@\text{nCN}_x$ catalyst was used. According to the authors' own cited reference (Nat. Synth. 2023, 2, 1092–1103), $\text{Ni}_1@\text{nCN}_x$ is expected to favor C–O coupling, which in this case would yield byproduct 4a. The authors must clearly state this and provide a mechanistic rationale for the observed difference in coupling selectivity (C–C vs. C–O) between the current semi-heterogeneous system and the previously reported heterogeneous system. Without such clarification, the mechanistic underpinnings of the semi-heterogeneous system—the core focus of the manuscript—remain ambiguous and must be addressed in the revised manuscript.

Comment 2:

While I acknowledge the practical limitations associated with photochemical scale-up, I remain unconvinced that the use of a numbering-up strategy in a batch system sufficiently demonstrates the scalability of the catalytic process. As discussed in reference 43 (Chem. Rev. 2022, 122, 2752–2906), numbering-up is a strategy more appropriately applied to continuous flow systems. In contrast, the current study employs a batch process, where this approach is less relevant. To substantiate claims of scalability, the authors should either demonstrate a genuine scale-up within a batch system or apply a numbering-up strategy in a flow system.

Comment 3:

The issue regarding yield comparison remains unresolved. The standard conditions in the present study require 72 hours at elevated temperature, whereas MacMillan's protocol achieves C–C coupling in 48 hours at room temperature. Under these markedly different conditions, similar yields (e.g., 71% in this study vs. 85% in MacMillan's work for compound 3a) do not constitute a valid comparison. Furthermore, the article cited by the authors (Chem. Sci. 2019, 10, 5837–5842) compares mechanochemical and solution-phase conditions within a single study, carefully controlling all variables aside from the activation method. In contrast, the current manuscript compares yields from two independent reports without such control, which invalidates the justification provided. To enable a sound comparison, the authors must present data obtained under matched experimental conditions. Without these corrections, I respectfully maintain that the justification for the manuscript remains unconvincing.

Version 3:

Reviewer comments:

Reviewer #4

(Remarks to the Author)

I have reviewed the revised manuscript and the responses to the reviewers' comments. The major issues have been adequately addressed, and the manuscript is now suitable for acceptance.

Point-by-point response to the Reviewers' comments

(original comments in blue, replies in black, actions in bold)

■ Reviewer #1

I evaluated the manuscript by Gianvito Vilé and co. entitled "Photocatalytic C(sp³)-C(sp³) cross-coupling of carboxylic acids and alkyl halides using a nickel complex and carbon nitride" with great interest. In the report, the author developed an approach merging nickel catalysis and heterogeneous photoredox catalysis, through the use carbon nitride nanosheets (nC₃N₄), to form a C(sp³)-C(sp³). The reaction proceeds via the use of carboxylic acid containing compounds (mostly cyclic 4, 5 and 6-membered rings) that undergo decarboxylation to form the corresponding alkyl radical the undergoes reaction with a halogenated substrate. Overall, the study is nicely executed but unfortunately, as stated by the authors, and as analyzed by this reviewer, the main advancement of the present study is the substitution of iridium photosensitizers for more abundant carbon nitride photosensitizers. Whereas this is a noble goal, since the reaction on some of these derivatives are already known and reported using Ir(III) photosensitizer, this study only focuses on repeating this transformation using another abundant photosensitizer. It is this reviewer's opinion that such iterative study should not warrant publication in Nature Communications as they solely rely on changing the photosensitizer. To emphasize this point, the mechanism as proposed relies on the oxidation of the proline derivative. As such, other organic photosensitizers such as acridinium and pyrylium derivatives could also be as efficient for this transformation and it is not conceivable to publish in Nature Communication each time a new photosensitizer is screened, unless additional mechanistic information are obtained, which is not the case here.

We sincerely thank the Referee for the opportunity to openly discuss this matter. While it may appear that our work simply substitutes a more abundant photosensitizer (carbon nitride) in place of the traditionally used Ir(III) photosensitizer, our study goes beyond a mere replacement and reflects a significant advancement in catalyst design and reaction engineering. The use of carbon nitride nanosheets (nC_N) leverages, in fact, a novel catalytic interface where nickel and nCN_x act cooperatively to achieve selective decarboxylative alkylation. This transformation required a fine-tuning of reaction conditions and a careful engineering of the catalyst interface to facilitate the desired coupling while avoiding side reactions. Through this design, we demonstrate that carbon nitride is not simply an alternative photosensitizer but a crucial component enabling controlled reactivity that was previously challenging to achieve with traditional photosensitizers. Moreover, modification of carbon nitride alters the selectivity patterns.

This aspect, perhaps not fully elaborated in the original submission, has now been addressed more comprehensively. We have added DFT calculations and spectroscopic studies, which enhance the mechanistic understanding of the reaction and provide valuable insights into the adaptive role of carbon nitride. These findings clarify how the carbon nitride interface facilitates the formation of decarboxylative C(sp³)-C(sp³) coupling, while its modification leads to ester formation under specific conditions, revealing the dynamic influence on reaction pathways.

We would like to add a final note: the field of chemistry requires not only the discovery of new reactions but also the development of greener catalysts—two complementary and equally important research directions. In this work, we address the latter by replacing a homogeneous, precious metal-based photosensitizer with an earth-abundant, heterogeneous alternative, aligning directly with green chemistry principles. By advancing the use of sustainable, metal-free photosensitizers, this research meets the pressing need for environmentally friendly and cost-effective catalytic systems that do not rely on scarce, expensive metals. The application of carbon nitride as a photosensitizer in our study not only cuts costs but also opens the door to broader applications in photoredox catalysis; and its scalability and environmental benefits provide a promising pathway for C–C bond formation, thus making a meaningful contribution to the need of more sustainable chemical production. Therefore, we are convinced that the work fits the scope of *Nature Communications*.

1) Figure 1: "High temeprature" should be "High temperature". The authors mention purple LED at 420 nm and then Blue LED at 420 nm (on page 7 and 8). This should be corrected. Table 1: The structure of the proline derivative is wrong (it should be COOH and not CH2OOH).

We have corrected these typographical errors. We extend our thanks to the Reviewer for their careful reading.

2) K-PHI is not defined in the text.

In the revised version of the manuscript, all photocatalysts (including K-PHI) have been defined. We thank the Reviewer for the comment.

3) The authors should emphasize also or compare the yields with those reported by MacMillan in 2016, which for 25% of them are roughly identical.

We agree with the Reviewer that this can be important. For the substrates that have been precedingly reported by MacMillan in 2016, **we now include a vis-à-vis comparison yield comparison.**

4) What is the recovery and recyclability of C3N4? The authors have a short paragraph on it that is immediately re-engaged in the same reaction but I wonder how much would be recovered at the end of the reaction and if it would operate as efficiently over 5 new reactions.

We thank the Reviewer for their suggestion. **We have now conducted 5 consecutive reactions**, consistently obtaining the alkylated product with yields consistently between 77% to 81%. After each reaction, the nCN_x catalyst was recovered by centrifugation, washed with acetonitrile and water, and dried at 65°C overnight. We monitored the recovery rate of the catalyst, obtaining quantitative yields in each instance. Although not requested, **we also carried out post-catalysis characterizations (i.e., N₂ physisorption, XRD, CHN elemental analysis, and ICP-OES)** to prove the structural robustness of the catalyst and the absence of any potential Ni doping of the carbon nitride carrier (ICP). These results validate the stability of the catalyst and its potential for long-term use in repeated catalytic cycles without degradation or contamination.

5) I could not judge the DFT calculations as it is beyond my area of expertise but I would recommend that the author carry out some excited-state quenching experiments to corroborate their mechanism (although it is identical to the one proposed by MacMillan). Stern-Volmer experiments using carbon nitride and the different partners (proline, Nickel, Proline and nickel together) should be investigated to investigate possible quenching routes.

We appreciate the Reviewer's suggestion. We would like to emphasize that the mechanism we propose differs from that of MacMillan's reported in *Nature* **2016**, 536–325. In MacMillan's mechanism, it was reported that Ni(0) first captures the decarboxylated radical to form Ni(I), followed by the oxidative addition of the alkyl bromide to generate Ni(III). In contrast, our proposed mechanism involves Ni(0) undergoing oxidative addition of the alkyl halide first, forming Ni(II), which is then followed by a radical oxidative trap, resulting in Ni(III). To gain insights into the SET step and validate the mechanism, we have conducted Stern-Volmer experiments to study the interaction of nCN_x with various substrates. Our results show significant quenching of nCN_x fluorescence upon interaction with deprotonated proline. **These new findings have been incorporated into the amended manuscript** to provide additional evidence supporting our proposed mechanism. We are grateful to the Reviewer for prompting this investigation.

■ Reviewer #2

sp³ C-sp³ C corss-coupling of carboxylic acids and alkyl halides presents a strong tool for the synthesis of a diverse array of functionalized drug molecules. This work modified the reported Ni-Ir process by replacing Ni with PCN, and the results are important and interesting in some degree.

We appreciate the Reviewer's recognition of the significance of these results and the broader impact they may have on the field of synthesis and process chemistry.

1) the roles of CN_x, Ni, ligand, base can be further explored.

We greatly appreciate the Reviewer's suggestion. In response, **we have conducted further experiments and DFT calculations to clarify the role of these compounds in the revised manuscript**. Specifically, Stern-Volmer quenching experiments have been performed and have shown a quenching interaction between the deprotonated carboxylic acid substrate and the photoexcited nCN_x, leading to a reduction in the fluorescence of the nCN_x photocatalyst. Furthermore, to better understand the base's selectivity, DFT calculations have been added to demonstrate the favorable interaction between the base counter anion and the deprotonated carboxylic acids, facilitating the decarboxylation step. Finally, electron paramagnetic resonance (EPR) experiments were performed to observe the presence of different oxidation states of Ni during the homogeneous catalytic cycle. These findings strengthen our mechanistic hypothesis and provide a comprehensive picture of the factors that contribute to the catalytic activity, selectivity, and efficiency of the system. **We have incorporated these new insights and experimental data into the revised manuscript** and hope that these additions address the Reviewer's suggestions satisfactorily. Thank you once again for your constructive feedback, which has greatly improved the clarity and depth of our study.

2) The yield of 3a increases, and that of 4a decreases as the Na₂CO₃ increases from 1 to 2 equiv; while the 3a yield decreases but the 4a yield increases as the Na₂CO₃ increases from 2 to 3; and then 3a increases but 4a decreases as the alkaline increases from 3 to 4. Please explain.

We thank the Reviewer for this observation regarding the yield variations of **3a** and **4a** with different Na₂CO₃ concentrations. While it is correct that there are small fluctuations in yield as the base concentration changes, these variations are minor, in the range of 5-6%. Given this limited change, we do not believe that strong conclusions can be drawn regarding the effect of Na₂CO₃ concentration on product distribution. Within the range of base concentrations and types investigated, we observed no significant influence on the reaction outcome.

3) Compare Table 8 and Table 1, there are too many same control results are repeatedly presented in the two tables.

Our intention was to provide a detailed description of the optimization procedures and control experiments in the Supporting Information (**Tables S1–S7**), while highlighting the key results in the main manuscript (**Table 1**). **In the revised manuscript, we have ensured clarity on this point.**

4) Many related reports regarding of sp³ C- sp³ C cross coupling are missed, ie. ACS Med. Chem. Lett. 2018, 9, 7, 773–777.

We appreciate the Reviewer's suggestion and **have included the referenced report** (*ACS Med. Chem. Lett.* **2018**, 9, 7, 773–777) for C(sp²)-C(sp³) coupling in the revised manuscript. Additionally, **we have incorporated other relevant photocatalytic reports regarding C(sp³)-C(sp³) bond formation**, including: *Adv. Synth. Catal.* **2020**, 362, 2367–2373; *J. Am. Chem. Soc.* **2018**, 140, 50, 17433–17438; and *J. Am. Chem. Soc.* **2023**, 145, 14, 7736–7742.

5) The description regarding of Figure 1 is different from the Figure. Figure 1a is not for a photocatalytic homogeneous decarboxylative C(sp²)-C(sp³) coupling using Ni catalyst and Ir photocatalyst; that for Figure 1b is also wrong.

We have corrected the typo. We extend our thanks to the Reviewer for their comment.

6) Where is the result for recyclability?

We thank the Reviewer for their question. In our revised manuscript, **we have repeated the reaction five times using nCN_x photocatalyst**, consistently achieving the alkylated product in the stable range of 77% to 81% yield. After each reaction, the nCN_x catalyst was recovered through centrifugation, washed with MeCN and water, and then dried in the oven overnight, resulting in a quantitative recovery of the catalyst in each instance. Additionally, post-catalysis characterization analysis was performed to demonstrate the stability and structure endurance of nCN_x in the five reaction cycles.

7) some experiemntal proof for the change of Ni chemical state is better to be presented?

We perceive this comment as related to the preceding discussion (point 1 above). Electron paramagnetic resonance (EPR) experiments were performed to observe the presence of different oxidation states of Ni during the homogeneous catalytic cycle. These findings align with and support the proposed mechanism.

8) Please explain how to obtain the compared CO₂ emission in Figure 5.

Many thanks for this valuable feedback. We have added further explanation on how the CO₂ emissions in Figure 5 were calculated to the revised manuscript. We hope this addition meets the expectations of the Reviewer.

■ Reviewer #3

This manuscript describes dual Ni/carbon nitride catalyzed C(sp³)-C(sp³) coupling of carboxylic acids and alkyl halides. Although replacing noble Ir by carbon nitride has been demonstrated previously, this study offers a more practical protocol for dicarboxylic C(sp³)-C(sp³) coupling, in terms of cost, catalyst recycle and low CO₂ emission.

We sincerely thank the Reviewer for their positive feedback.

(1) In an earlier publication, the authors reported carbon-oxygen coupling from the same coupling partners employing single atom Ni anchored on carbon nitride. Carbon-oxygen coupling also proceed with different yield, depending on conditions. It is worth to find the underlying reasons for different coupling manners, which will increase the scientific impact and guide catalyst development.

The formation of compounds **3a** and **4a** proceeds via distinct reaction pathways. Therefore, when the pathway leading to **3a** is slowed or inhibited, the formation of **4a** becomes more favorable. We propose that the selectivity for **3a** is linked to a preferred single electron transfer (SET) step, which ultimately leads to the decarboxylation process. Among the factors promoting SET, we emphasize the use of 420 nm light over 460 nm, which corresponds to the maximal absorbance of carbon nitride. This may help stabilize its triplet state. Furthermore, in acetonitrile, a polar solvent known for facilitating SET, the electron transfer from photoexcited nCN_x to the deprotonated carboxylic acid becomes the dominant pathway. Finally, a selectivity trend was observed between the formation of **3a** and the type of base utilized. To stress this point, Stern-Vomer experiments indicate a stronger interaction of the photoexcited nCN_x between **1** in presence of Na₂CO₃ than in presence of Cs₂CO₃. Besides, **we modeled the interactions between Na, K, and DBU with the deprotonated carboxylic acid by considering the Boc-protected L-proline**. The results suggest that Na may better promote the decarboxylation process while the DBU will block it (see Figure 1). Collectively, these results shed light on the factors that govern the interplay between carbon-oxygen and carbon-carbon coupling, and expand the versatility of single-atom catalysts for diverse coupling reactions. **We have included this discussion in the text.**

Figure 1. Base effect on the decarboxylation process.

(2) The statement “molecular insights into the role of nickel single atoms in facilitating photodecarboxylation and subsequent C–C bond formation”, where single atom is used for Ni in the bpy coordinated complex might not be suitable.

We thank to the Reviewer’s for their comment. **In the revised manuscript, we have corrected that statement.**

(3) The manuscript requires substantial reorganization. The characterizations and condition optimizations should be condensed, and many details are better suited for inclusion in the Supplementary Materials.

We are not in line with the Reviewer’s suggestion to condense the characterization and condition optimization sections, as these elements are integral to our study’s focus on catalyst design. Detailed descriptions of these aspects provide critical insights into how specific conditions and structural features influence catalytic performance. Since we aim to offer a comprehensive understanding that will be valuable for researchers working on similar catalyst development, we have not taken any action on this specific matter.

(4) The photoreaction conditions should be provided in detail (solution volume, incident light intensity at different wavelengths); MeCN (x M) is confusing, whose concentration in the bracket indicate? MeCN or alkyl bromide?

We appreciate the Reviewer’s suggestion. **In the revised version of Supplementary Information, we now better explain the reaction conditions**, indicating the reaction volume and the light intensity.

We apologize if the concentration of MeCN was confusing; this was related to the limiting reagent, which is the alkyl halide (0.2 mmol, 1.0 equiv.). therefore, $0.05\text{M} = 0.2/V$; $V = 4\text{ mL}$. **In the revised version of the manuscript we express the amount of MeCN in mL instead of M.**

(5) Please provide the data for catalytic performance of the recovered catalyst, together with the structural characterizations (XRD, TEM, and the possible Ni aggregates)

We thank the Reviewer for their suggestion. In our revised manuscript, **we have carried out post-catalysis characterizations** (BET, XRD, elemental analysis) **to prove the structural robustness of the catalyst.**

(6) Ni(0) is proposed to interact with alkyl halide or Intermediate IV. The initial state of Ni is Ni(II). And there are controversies on the active forms of Ni (Ni(I) or Ni(0)). The authors may want to track the evolution of Ni on CN experimentally.

We agree that tracking the evolution of nickel during the catalytic cycle is essential. While the initial state of nickel in our system is indeed Ni(II), electron paramagnetic resonance (EPR) experiments confirm the presence of distinct oxidation states of nickel throughout the catalytic cycle. These EPR findings align well with the proposed mechanism.

(7) Beside free energy changes, the transition states and the corresponding barriers in the two proposed pathways should be provided to support the assignment of the dominant one.

We have modified the text to better explain our choice. We considered the analysis of the reaction barriers using the thermodynamics approach developed by Nørskov *et al.* (see: *J. Catal.* **2002**, *209*, 275-278; *J. Catal.* **2001**, *197*, 229-231; *Nat. Chem.* **2009**, *1*, 37-46; *J. Catal.* **2004**, *224*, 206-217; *Nat. Mater.* **2006**, *5*, 909-913; *Adv. Catal.* **2000**, *45*, 71-129) and assuming the existence of relations correlating the activation and free energies (see: *J. Catal.* **2002**, *209*, 275-278; *J. Catal.* **2001**, *197*, 229-231; *Nat. Chem.* **2009**, *1*, 37-46; *J. Catal.* **2004**, *224*, 206-217; *Nat. Mater.* **2006**, *5*, 909-913; *Adv. Catal.* **2000**, *45*, 71-129).

According to the Bronsted-Evans-Polanyi relationship (BEP), the more exergonic a reaction is, the lower the barrier will be, and the transition state energies should scale with the free energies (see: *Chem. Rev.* **1928**, *5*, 231-338; *Trans. Faraday Soc.* **1938**, *34*, 11-24; *J. Catal.* **2002**, *209*, 275-278). However, calculating the transition states would introduce errors due to the inherent difficulty of accounting for multiple reaction pathways and the solvation effects of leaving groups like Br, which to date are hard to capture thermodynamically in transition states. In light of the potential inaccuracies, we have chosen to avoid an extensive discussion on this aspect to preserve the scientific consistency and reliability of our DFT profile and energy levels.

(8) Calculation and references for GHG emissions should be provided

We have included a more thorough explanation in the revised manuscript regarding the GHG emissions. We trust that this addition enhances the clarity of our work.

■ Reviewer #4

In this manuscript, Gianvito Vilé *et al.* present a study on C(sp³)-C(sp³) cross-coupling in a semi-heterogeneous system using an nCN_x photocatalyst. While the exploration of such catalysts and their application in cross-coupling reactions is undoubtedly significant, and the concept of utilizing nCN_x as a heterogeneous photocatalyst is intriguing, the manuscript falls short in distinguishing this approach from prior research. For instance, in a previous study (*Nat. Synth.* **2023**, *2*, 1092-1103), Gianvito Vilé already employed a Ni single-atom catalyst for C-O coupling but did not adequately elucidate the rationale behind the occurrence of decarboxylation in this study. A more in-depth investigation into the decarboxylation mechanism is needed, particularly in light of earlier reports (*Angew. Chem. Int. Ed.* **2024**, *63*, e202405902; *Adv. Synth. Catal.* **2020**, *362*, 3898-3904). Additionally, it is well-established that homogeneous nickel catalysts are prone to deactivation in dual photoredox/nickel-catalyzed processes, as highlighted in a previous study (*Nat. Catal.* **2020**, *3*, 611-620). This manuscript does not sufficiently address this critical issue.

We sincerely appreciate the Reviewer's insightful comment and the recognition of the significance of our work on C(sp³)-C(sp³) cross-coupling using an nCN_x photocatalyst. However, we understand the Reviewer's concern regarding the need to better differentiate this approach from prior research. To address this, the revised manuscript emphasizes the unique aspects of the semi-heterogeneous system, where the nCN_x photocatalyst operates in a distinct manner compared to conventional homogeneous systems. The manuscript highlights the advantages of this dual-phase approach, including improved catalyst stability and recyclability, which were not explored in the context of cross-coupling reactions in prior studies. In response to the Reviewer's suggestion for a more detailed investigation of the decarboxylation mechanism, we have now included additional experiments and computational studies. These include Stern-Volmer quenching experiments, DFT calculations, and EPR measurements, which provide a clearer understanding of the reaction pathway and strengthen the explanation of the decarboxylation step.

Regarding the Reviewer's comment on the potential deactivation of homogeneous nickel catalysts in dual photoredox/nickel-catalyzed processes, we acknowledge the importance of this issue. The manuscript now explicitly addresses how the semi-heterogeneous system helps mitigate catalyst deactivation through the stable interaction between the photocatalyst and the nickel species, providing enhanced stability and overcoming the typical deactivation challenges observed in homogeneous systems.

We trust that these additions and clarifications will meet the Reviewer's expectations and significantly improve the manuscript.

The previous studies referenced on page 2 do not correspond accurately with the information presented in Figure 1. For instance, the study shown in Figure 1c is from 2016, not 2014 as stated in the text. Please ensure that all studies mentioned in the text are correctly aligned with those depicted in the figure, and address any typographical errors.

We extend our thanks to the Reviewer for noting these errors. **We have corrected all typographical mistakes in the amended text.**

The authors optimized several photocatalysts, including nCN_x, gCN_x, mpg-CN_x, K-PHI, and recovered nCN. However, the manuscript lacks sufficient characterization data to substantiate the comparison and validation of nCN_x against the other photocatalysts. Although nCN_x was prepared through a thermal exfoliation process from gCN_x, no data are provided to describe the morphology, thickness, or surface area of nCN_x relative to gCN_x. Therefore, it is essential to include comprehensive characterization data for all the photocatalysts in Figure 2, such as BET, XRD, AFM, and UV analyses. Additionally, TEM, SEM, XPS, and FT-IR data should be provided for each photocatalyst. Furthermore, the method used to calculate the reported C/N ratios ('0.61-0.67') should be clearly explained.

We thank the Reviewer for their suggestion. In our revised manuscript, **all the photocatalysts tested have been defined and characterized.** We have also added missing details in the text.

In this study, the authors present an optimization of a photocatalyst, identifying nCN_x as the optimal choice. However, several concerns arise regarding the rationale behind selecting nCN_x over other catalysts. The authors suggest that mpg-CN_x is less effective due to its higher solution viscosity, which they claim reduces the availability of active sites. This assertion, however, lacks sufficient experimental evidence or comprehensive characterization of the photocatalysts in question. While it is acknowledged that mpg-CN_x may exhibit increased viscosity, this factor alone does not provide a strong basis for dismissing its catalytic potential. In fact, mpg-CN_x typically has a larger surface area, which is generally associated with an increased number of active sites, as documented in the literature (Chem. Eur. J. 2015, 21, 526–530; J. Phys. Chem. C 2012, 116, 19644–19652; Angew. Chem. Int. Ed. 2015, 54, 12868–12884). This contradicts the authors' claim that mpg-CN_x is less suitable due to fewer active sites. The manuscript lacks detailed characterization data, such as BET surface measurements, pore size distribution, or SEM/TEM imaging, which are essential for substantiating the claim that mpg-CN_x has a reduced number of active sites or is significantly impacted by viscosity-related issues. Moreover, the authors report comparable reaction yields between CN_x (3a 41% and 4a 23%) and mpg-CN_x (3a 34% and 4a 18%), indicating that mpg-CN_x could be a viable alternative. The rejection of its efficacy based on unproven assumptions about solution viscosity and active site availability is unconvincing.

We sincerely appreciate the Reviewer's insightful suggestion and completely agree that these are important considerations. We acknowledge that mpgCN_x typically exhibits a larger surface area, which, as the Reviewer correctly points out, is generally associated with an increased number of active sites. However, during the course of our experimental campaign, we noticed that mpgCN_x formed a more viscous reaction environment and, moreover, it adhered to the walls, which potentially led to less efficient light irradiation. While we acknowledge that the reactivity of both compounds is comparable, we chose to proceed with optimization using nCN_x due to its easier reaction. However, in the revised paper, we removed any claims that mpgCN_x is less effective and we have revised the manuscript to present a more balanced discussion, emphasizing

that while nCN_x was found to be optimal for our specific conditions, mpgCN_x remains a viable candidate for future studies. We hope these revisions clarify our approach and address the Reviewer's concerns.

In addition to the concerns mentioned above, there is an inconsistency in the manuscript regarding the mention of "boron-doped mpg-CN_x." This material is introduced without prior context or discussion in the manuscript, leading to confusion. The authors should ensure that all materials discussed are properly introduced and contextualized.

We thank to the Reviewer's for their valuable comment. We fully agree that the statement needed correction. **In the revised manuscript, we have removed any mention of using a boron-doped catalyst.**

The manuscript includes product yield data that have been confirmed using HPLC calibration. However, the authors have not provided the detailed evidence of the HPLC calibration curves that were used. To ensure scientific reproducibility, it is imperative that the authors provide comprehensive details regarding the calibration process. Furthermore, there are notable inconsistencies between the product yield values reported in the main text and those presented in the Supporting Information (SI). These inconsistencies call into question the credibility of the data and must be addressed. The authors should conduct a thorough review and make the necessary corrections to ensure consistency across the manuscript. Additionally, the SI lacks NMR spectra for several compounds discussed, despite the fact that these are typically required to confirm the synthesized products. The authors must provide the NMR data for all relevant compounds to support the claims made in the manuscript.

We thank the Reviewer's for their valuable suggestions. We concur with the Reviewer's comments. **In the revised version of the manuscript we provided details of the HPLC calibration curve.** Upon purifying the product **3a**, several aliquots of 100 microliters were taken, resembling different reaction yields (0, 12,5%, 25%, 50%, 75% and 100%), The different absorbances were measured, obtaining a robust linear relationship between concentration and AUC ($r^2 = 0.997$).

Additionally, we agree that the NMR data was missing. **In the revised manuscript we incorporated ¹H-NMR, ¹³C-NMR and ¹⁹F-NMR (if needed) of every purified product.**

This paper presents DFT calculation data to explain the Ni-catalyzed reaction pathway. However, the DFT data presented, which uses proline, does not offer a sufficiently reliable explanation for the mechanistic pathway. Based on the substrate scope study in this paper, the reactivity and selectivity are influenced by the substituents, with even methyl-substituted proline showing no reactivity. Therefore, it would be more appropriate to present DFT calculation results using Boc-protected proline rather than proline. Additionally, the Ni-catalytic cycle has already been extensively addressed in previous studies (*Nature* 2016, 536, 322–325; *J. Org. Chem.* 2024, 89, 11136–11147; *Science* 2014, 345, 437–440). It would be beneficial to develop data that provides deeper mechanistic insights into decarboxylation beyond esterification.

We appreciate the Reviewer's comment on the DFT part. The suggested papers (*Nature* 2016, 536, 322–325; *J. Org. Chem.* 2024, 89, 11136–11147; *Science* 2014, 345, 437–440) have been included in the manuscript.

We also agree with the Reviewer's point on the Boc-protected proline; we have thus addressed the point by calculating the full reaction profile in the presence of the Boc group. As shown in the amended paper, the reaction energy is equivalent with and without Boc, although more stable intermediates are formed when comparing Boc-protected and free proline. This result is not surprising. In fact, while the Boc group serves as a protective group in proline to prevent its amine functionality from unwanted reactions, influencing the local environment of the reactant and intermediate states, the fact that DFT calculations did not show any significant change in the reaction profile is due to the inherent nature of DFT as a local optimization method. DFT tends to capture the electronic structure accurately but may not fully account for long-range effects or subtle changes that could arise from the conformational flexibility or dynamic interactions of the Boc group in a real reaction context. Therefore, the overall reaction mechanism remains energetically equivalent in both cases, which is consistent with the predictions from DFT. As a result, all our mechanistic conclusions are

unchanged, and, for example, the Ni(II) intermediate remains still more stable than Ni(I), as shown below in Figure 2 where the energetic profile with proline (a) and the Boc-protected proline (b) are shown.

Figure 2. Energetic profile for the C-C coupling by considering the (a) proline molecule and the (b) Boc-protected proline.

The authors present GHG emission data derived from the equations described in the 'Energy Calculations' section of the Supplementary Information (SI) to support the sustainability of carbon nitride compared to Ir-based complexes. However, the mentioned 'Energy Calculation' section is absent from the SI. To substantiate the sustainability advantages of carbon nitride, it is recommended that the authors provide a detailed calculation process along with reliable references.

Thank you and we sincerely apologize that this section was missing in the original submission. In response, we have added a detailed explanation on how the CO₂ emissions in Figure 5 were calculated. We hope this addition aligns well with the Reviewer's expectations and provides further clarity.

■ Reviewer #5

The authors described the use of graphitic carbon nitride as a photocatalytic system to drive cross-coupling between alkyl halides and carboxylic acids. Mechanistic studies are also reported. In my view, the manuscript has been well written and the results are sound. Thus, I recommend its publication after having addressed the points below.

We thank the Reviewer for recognizing the clarity and robustness of our results. We will carefully address the points they raised to ensure the manuscript meets the highest standards for publication.

1. Additional references regarding the use of carbon nitride-based photocatalytic systems for the functionalization of organic compounds should be included in the introductory section, for instance: *Science* 365, 360–366 (2019), *Angew. Chem. Int. Ed.*, 2023, e202313540, *ACS Catal.* 2023, 13, 13414–13422, *Sci. Adv.*, 2020, 6, eabc9923, *ACS Catal.* 2024, 14, 11308–11317, *ACS Nano*, 2021, 15, 3621–3630, *Chem. Sci.*, 2022, 13, 9927, *Nature Catalysis*, 3, 611–620 (2020), *Adv. Sci.* 2023, 10, 2303781, among others.

We thank the Reviewer for their helpful suggestion regarding the inclusion of additional references on carbon nitride-based photocatalytic systems. We agree that these references would provide a more balanced and comprehensive view of the field. We have thus cited these works in the introductory section of the manuscript.

2. The reaction scope should be expanded. Is it possible to use fatty acids, alkyl chlorides and alkyl triflates as starting materials? Moreover, the generality of the photocatalytic system with respect to more synthetically useful organic substrates, namely natural products or active drugs, should be addressed. For instance, is it possible to use other natural products or bio-active molecules containing carboxylic moieties as substrates?

We thank the Reviewer for this valuable suggestion. In line with their recommendation, **we have expanded the reaction scope demonstrating the functionalization of bioactive compounds, including lysine and levodopa precursors.** These new results validate the versatility and applicability of our photocatalytic system to a wide range of synthetically relevant substrates, supporting its potential for broader utility in organic synthesis.

3. Is it possible to perform the model reaction in a gram scale?

We thank the Reviewer for this insightful question. Scalability can be approached through either *sizing up* (increasing the volume of a single reaction vessel) or *numbering up* (using multiple reaction vessels in parallel). However, sizing up is limited in photocatalytic systems due to the constraints imposed by the Lambert-Beer Law, which describes the exponential attenuation of light as it penetrates through an absorbing medium. According to this, light intensity decreases with path length and concentration of the photocatalyst, resulting in uneven photon distribution and suboptimal reaction rates in larger reaction volumes. Consequently, as vessel size increases, light penetration becomes insufficient to drive the reaction uniformly, leading to decreased efficiency and yield.

To overcome this limitation and address the Reviewer's concern, we adopted the *numbering up* strategy, where multiple smaller reaction vessels are used in parallel, each receiving adequate and consistent light exposure to maintain reaction efficiency. This approach aligns well with the guidelines developed by several scientists who have pioneered scale up/scale down approaches for process intensification (*Chem. Eng. Process.* **2007**, *46*, 781-789; *Ind. Eng. Chem. Res.* **2019**, *58*, 5349-5357; *Chem. Eng. Sci. X* **2021**, *10*, 100097). Applying this approach, **we successfully conducted the model reaction in a single pot and simultaneously with 15 parallel reactions. This enabled us to scale up the reaction from 0.2mmol of reagent (71% isolated yield of desired product 3a) to 3.0 mmol of reagent (77% isolated yield of desired product 3a).** This demonstrates the practicality and effectiveness of numbering up for scaling photocatalytic batch reactions, allowing for higher throughput without compromising yield.

4. I suggest to include a general procedure for the photocatalytic experiments along with a picture of the reaction set-up and its description within the SI.

We have included a picture of the photoreactor in the Supporting Information.

5. The authors should better characterize the photocatalyst after a series of catalytic cycles.

We thank the Reviewer for their suggestion. In response, **we have included a recyclability test for the nCN_x photocatalyst in the revised manuscript.** Specifically, we conducted five consecutive reactions, consistently obtaining the alkylated product with yields consistently between 77% to 81%. After each reaction, the nCN_x catalyst was recovered by centrifugation, washed with acetonitrile and water, and dried at 65°C overnight. We monitored the recovery rate of the catalyst, obtaining quantitative (95-99%) yields in each instance. **We carried out post-catalysis characterizations (i.e., N₂ physisorption, XRD, CHN elemental analysis, and ICP-OES)** to prove the structural robustness of the catalyst and the absence of any potential Ni doping of the carbon nitride carrier (ICP).

6. I suggest an improvement on the structure of the manuscript due to the presence of typos (e.g., structure compound 1 within table 1).

We have corrected all the typographical errors. We extend our thanks to the Reviewer for their careful reading.

Point-by-point response to the Reviewers' comments

(original comments in blue, replies in black, actions in bold)

■ Reviewer #1

I commend the authors for the quality and seriousness of their revisions. The manuscript is much improved and I felt like the author responded professionally to all the comments. I have a very few minor comments:

We appreciate the Reviewer's positive feedback. We are also pleased that our responses were perceived as professional. The Reviewer's comments have been instrumental in refining the manuscript, and we have now addressed the remaining minor points herein.

In the novel figure 4, panel B, the "micro" symbol does not display properly, and only "u" is readable.

We thank the Reviewer for pointing out the typo. **We have corrected the symbol** to ensure that μ is clearly readable.

In Figure 4, panel C, the authors mislabeled the Y-axis, and I should be I₀/I and not I/I₀. In addition, compound 2, 1 and 1+ Nicat + dMeObpy (probably mislabeled as well), seem to have a negative slope that the author did not comment on. A negative slope would imply that the photoluminescence intensity increases as the quencher is added.

We are grateful for the Reviewer's insight. **We have adjusted the label and the slope accordingly.**

Still in figure 4, I would recommend forcing the intercept of all lines at (0;1), as this is a "true" datapoint according to the Stern-Volmer equation, $I_0/I = 1 + K_{sv}[Q]$, so when $[Q] = 0$, I_0/I should be 1.

Thank you. **We have corrected Figure 4 in the revised manuscript.**

■ Reviewer #2

I read through the revised draft and the response to comments carefully, and find that the issues were basically addressed, and now can be acceptable.

We thank the Reviewer for their positive feedback and are pleased that the revised version of the manuscript is considered suitable for publication in *Nature Communications*.

■ Reviewer #3

In the revised manuscript, the authors provided more insights into the C-C coupling (other than the previously reported C-O) by systematically excited state quenching, photon energy/base/solvent dependence and calculation. Other issues are also addressed.

We appreciate the Reviewer's recognition of our efforts to provide deeper insights into C-C coupling, and we are grateful for the constructive feedback received.

■ Reviewer #4

In the revised manuscript, Gianvito Vilé and colleagues have addressed some of the issues raised in previous comments. While the updated version demonstrates improvements, several critical concerns remain, rendering the manuscript unsuitable for publication in *Nature Communications*. The manuscript still lacks a thorough characterization of the materials used, which is essential for validating the reported material properties. Moreover, the choice of a semi-heterogeneous system requires a stronger justification to establish the significance of the findings. Additional concerns stem from experimental aspects, particularly the

"numbering-up" scale-up process, which raises doubts about the reliability and scalability of the results. Consequently, despite the authors' notable efforts, I regret to recommend against publication in *Nature Communications*. Detailed comments are provided below:

We sincerely appreciate the Reviewer's time and effort in highlighting areas for improvement. However, we respectfully disagree with the Reviewer's assessment and we are convinced that this study is well-suited for *Nature Communications*. Nonetheless, we remain committed to improving the quality and clarity of our work and have made further changes to refine our approach and ensure a more rigorous presentation of our findings.

Comment #1: The manuscript does not sufficiently highlight the advantages of the semi-heterogeneous system through a detailed comparison with other catalytic systems, such as homogeneous and single-atom heterogeneous systems. While the authors mention that the semi-heterogeneous system mitigates catalyst deactivation via stable interactions between the photocatalyst and nickel species, the manuscript lacks a clear and rational explanation to support this claim.

While we appreciate the Reviewer's feedback, we respectfully disagree with them. The advantages of the semi-heterogeneous system have been clearly demonstrated through recycling tests, economic comparisons, and an in-depth analysis of GHG emissions. To address this point once again, **we have now performed a direct comparison of reaction outcomes between the reported Ir complex homogeneous photocatalyst, our conditions, and a fully heterogeneous Ni₁@nCN_x system.** This Ni₁@nCN_x single-atom catalyst was recently synthesized and characterized by our group (see *Nat. Synth.* **2023**, 2, 1092-1103). The results highlight the advantages of the semi-heterogeneous approach we have developed.

Entry	Conditions	Yield of 3a
1	NiCl ₂ ·glyme and Ir[dF(CF ₃)ppy] ₂ (dtbbpy)PF ₆	61%
2	NiCl ₂ ·glyme and nCN _x	75%
3	Ni@nCN _x	3%

Specifically, the table shows that, under optimized conditions, the semi-heterogeneous system surpasses not only the homogeneous catalysts, but also the heterogeneous single-atom catalysis. We attribute this to the stringent environmental requirements of Ni for catalytic activity, which are hindered by the absence of suitable ligands necessary for oxidative addition over alkyl bromides. Furthermore, since Ni in the single-atom catalyst is not present in the same phase as the substrate, the interactions between the catalyst and substrates are significantly reduced. However, we want to point out that we cannot exclude that, through catalysis engineering, a fully heterogeneous system may be developed in the near future. Upon discussing this with our co-authors, we decided to exclude the single-atom catalyst results from the manuscript, as they fall outside the scope of our study.

Furthermore, we want to remark that, in our studies, we propose a synergistic catalytic mechanism involving two distinct catalytic cycles operating simultaneously, and we never state that the stable interaction between nCN_x and nickel mitigates catalyst deactivation.

Comment #2: Although the authors stated that they included additional characterizations, critical analyses such as TEM, SEM, XPS, and AFM remain missing. The explanation of catalyst screening results is also insufficient to justify the selection of nCN_x as the optimal material. Characterization data for the recovered nCN_x are particularly lacking. For instance, TEM images and EDS mapping are necessary to determine

whether Ni aggregates have formed. Despite claims that such data were included, they are absent from both the manuscript and the supplementary information.

We thank the Reviewer for the feedback. **In the revised manuscript, we have further characterized the material, including, SEM, XPS and TEM for the fresh and recycled nCN_x (Figure S3-5).** The results are in line with the literature, showing that the structural and electronic properties of the material remain consistent with previously reported findings. Given that carbon nitride is a well-studied material with extensive characterization available in the literature (see *ACS Sustainable Chem. Eng.* **2023**, 11, 5284–5292; *ACS Appl. Nano Mater.* **2022**, 5, 14520–14528; *J. Colloid. Interface Sci.* **2024**, 673, 943-957; *ACS Catal.* **2021**, 11, 1593–1603; *ACS Omega* **2019**, 4, 12544–12554), we do not consider it necessary to include redundant analyses such as AFM, which would not provide significantly novel insights.

Moreover, AFM microscopes are currently not available at our institution, and conducting these analyses would require involving additional authors from outside our institution who, in our view, do not meet the criteria for authorship, as the analyses are not fundamental to the study.

In terms of stability, we have provided sufficient evidence demonstrating the robustness of the material, as shown in **Figure S3** and **Figure S5**. After five catalytic cycles, the presence of Ni adhered to nCN_x photocatalyst was consistent with the findings of König and Noël (see *ACS Catal.* **2021**, 11, 3, 1593–1603; *Angew. Chem. Int. Ed.* **2024**, 63, e202405902). The recycled photocatalyst was found to be inactive without the addition of the Ni homogeneous catalyst (**Table S9, entry 6**), and washing away the adhered Ni traces from nCN_x with 1 M HCl did not impact its intrinsic photocatalytic activity (**Table S9, entry 7**). In this context, we are convinced that no further characterization is considered necessary, as it can be found in *Angew. Chem. Int. Ed.* **2024**, 63, e202405902.

Comment #3: For gram-scale synthesis, the authors employed a numbering-up strategy instead of scaling up the system size. This approach is unconvincing for demonstrating the scalability of heterogeneous semiconductors. Moreover, the numbering-up experiments were conducted on an even smaller scale (0.2 mmol) than the optimized conditions (0.3 mmol). As a result, the manuscript does not sufficiently support the claim that nCN_x is a sustainable, efficient, and cost-effective alternative to traditional iridium-based photocatalysts, especially from an industrial perspective.

We encourage the Reviewer to carefully revisit the manuscript. The reaction scale for optimization, scope, and scale-up studies consistently employed 0.2 mmol of alkyl bromide (**2**), and 0.3 mmol of carboxylic acid (**1**), and this can be easily verified for the scale-up experiment as well. In fact, the scale-up was performed using 15 vials and a total of 3 mmol of alkyl bromide (our limiting reagent); therefore, each vial contained 0.2 mmol of alkyl bromide (3 mmol / 15 vials = 0.2 mmol).

Besides, as the Reviewer knows, one major limitation of photocatalysis is the challenge of scalability due to light absorption constraints. This issue arises from the attenuation effect of photon transport, as described by the Lambert-Beer law, which prevents the conventional scale-up of batch reactions by simply increasing reactor dimensions. In large reactors, the distribution of heterogeneous catalyst can be inefficient, leading to underutilization in certain regions. A numbering-up strategy addresses these challenges by optimizing the surface area and ensuring a more effective contact between the light and the photocatalyst. This approach enhances photon penetration, improving overall reaction efficiency while reducing energy costs.

We also respectfully disagree with the Reviewer's final statement. Iridium-based homogeneous photocatalysts often present challenges for large-scale applications due to their high cost, limited availability, and difficulties in recyclability. Our study demonstrates that solid nCN_x exhibits catalytic activity comparable to its homogeneous counterpart while providing key advantages, including lower cost, greater abundance, and enhanced recyclability. These characteristics make nCN_x a promising alternative for large scale decarboxylative cross-coupling reactions.

Comment #4: The authors propose that energy differences resulting from interactions between 1 and the base explain variations in product yields. This rationale is plausible in cases without a base (1.48 eV) or with DBU (2.31 eV). However, the manuscript fails to address the significant discrepancies between Na₂CO₃ and K₂CO₃ results. Despite their similar intermediate energies (1.24 eV for Na₂CO₃ and 1.25 eV for K₂CO₃),

their yields differ substantially, with Na₂CO₃ yielding 33% of **3a** and 17% of **4a**, while K₂CO₃ yields 34% and 44%, respectively. Additional justification or experimental data is needed to clarify these differences.

We thank the Referee for his comment. The interaction of sodium and potassium cations from sodium and potassium carbonates with deprotonated Boc-L-proline leads to the release of radical Boc-proline and CO₂, with similar energy barriers for each cation: 1.24 eV for Na₂CO₃ and 1.25 eV for K₂CO₃. However, the sodium cation forms an intermediate that has a stronger and more localized basic environment compared to that of the potassium-based intermediate. This effect is primarily attributed to the smaller size of the sodium cation and its electronic interactions with the deprotonated Boc-L-proline. The sodium cation enhances the stabilization of the intermediate by forming two bonds with oxygen atoms, each measuring approximately 2.21 Å in length (**Figure 1a**). This strong interaction favors the selective formation of product **3a** over byproduct **4a**. In contrast, the potassium cation forms an intermediate with three bonds to oxygen atoms: two at 2.59 Å and one at 2.70 Å (**Figure 1b**). These weaker interactions do not provide the same stabilizing effect, resulting in the formation of both product **3a** and byproduct **4a** at comparable rates. **This discussion has now been included in the revised manuscript.**

Figure 1. Effect of the decarboxylation process in the presence of sodium (a) and potassium (b) cations from inorganic bases.

Comment #5: Previously, it was suggested to provide detailed descriptions of the “Energy Calculations” for GHG emission data. However, the revised manuscript still lacks a clear explanation of the calculation procedures and data references. To emphasize the environmental advantages of the semi-heterogeneous system over homogeneous systems, a detailed explanation of how the GHG emission data were derived is essential.

We understand that LCA assessment is not a common practice for organic and inorganic chemists, but we want to assure the Reviewer that all necessary details are provided in the Supporting Information. The study followed a cradle-to-gate approach, meaning that it considers emissions from raw material extraction, acquisition, and production, while excluding usage and disposal stages. The functional unit was defined as 1 kg of the target compound. The life cycle inventory was performed collecting all relevant data on material and energy inputs, emissions, and waste flows associated with the catalytic reaction, and calculating the emissions accordingly. To do this, the reaction was inserted LCA software, allowing for the quantification of energy requirements and mass balances with the experimental information required. The background life cycle inventories for raw materials and energy sources were sourced in Ecoinvent 3.9.1 cut-off with Carbon Minds databases. The life cycle impact assessment phase was conducted translating raw emissions data into meaningful environmental indicators. In this study, the Global Warming Potential metric was used, which measures greenhouse gas emissions in kg CO₂ equivalent over a 100-year time horizon, following the methodology outlined in the 6th IPCC assessment report. These steps are common practice in the chemical engineering field, and our group has already published similar LCA studies that have received recognition from our peers (see *ACS Sustainable Chem. Eng.* **2025**, 13, 7, 2864–2874; *Cell Rep.* **2025**, 2, 1, 100286).

Thus, we are confident that the LCA calculations are robust and sufficiently detailed to highlight the environmental advantages of the semi-heterogeneous system over homogeneous alternatives.

Comment #6: Several inconsistencies in the manuscript undermine its reliability. These include: (1) Comparison of substrate yield with a prior study (MacMillan, 2016), despite differing experimental conditions (Ir system: room temperature, 48 h), which makes the comparison invalid. (2) Unexplained changes in nCN_x surface area from 12 to 23 m²/g, raising questions about synthesis reproducibility. (3) Conflicting details about the washing procedure for the recovered catalyst. The supporting information states the catalyst was washed with ethyl acetate and water, while the manuscript and prior responses mention acetonitrile.

We appreciate the Reviewer's perspective and insights. The comparison was included at the request of the Editor and another Reviewer. Comparing substrate yield is a valid approach, as demonstrated by numerous examples in the literature where reactions under different conditions have been compared (e.g., see *Chem. Sci.* **2019**, 10, 5837–5842, where mechanochemistry vs. solution-based reactions have been compared).

Regarding the surface area of nCN_x, we apologize for the oversight, and we confirm the value of 23 m²/g. To further substantiate this, we prepared three additional batches, consistently obtaining values between 21 and 24 m²/g. Thus, we can now determine that, by using our synthetic recipe, the reader will obtain a surface area of 22.5±1.5 m²/g. The value previously-included in the very first version of this manuscript was an error, as we erroneously utilized the surface area of a gCN_x batch synthesized with a heating ramp of 5 °C min⁻¹. We apologize for the oversight. We also thank the Reviewer for the (correct) observation regarding the solvent choice for washing the photocatalyst. Both acetonitrile (MeCN) and ethyl acetate (EtOAc) are indeed suitable solvents for washing the nCN_x photocatalyst, as they both effectively remove residual organic reagents and byproducts. However, we opted for EtOAc in our studies as it is considered a greener solvent and has broader industrial acceptance compared to MeCN (see *Green Chem.* **2016**, 18, 3879–3890). **We have corrected the typographical error in the revised manuscript.**

Comment #7: Several minor typos and formatting errors persist in the manuscript. For instance, the author's name is misspelled as "MCMillan" instead of "MacMillan" on page 3. Additionally, discrepancies between figure and data references within the manuscript need to be corrected.

We thank the Reviewer for pointing out these typos. Additionally, we have performed thorough proofreading to ensure consistency and accuracy throughout the text, and corrected all remaining errors.

■ Reviewer #5

The revised version of the Manuscript has been well organized and the results are very interesting. I personally found this version of the manuscript more detailed and clear. In addition, the authors addressed most of the reviewers' comments satisfactory. Thus, I think that the new version of the manuscript is now suitable for publication in Nature Communications.

We thank the Reviewer for their encouraging feedback and for recognizing the improvements in the clarity and organization of the manuscript.

Point-by-point response to the Reviewers' comments

(original comments in blue, replies in black, actions in bold)

■ Reviewer #1

I was personally satisfied by the changes carried out last time by the authors. I have however to stress that I agree with reviewer 4, that a "numbering-up" approach is, in my opinion, not a proper scale up, but rather represents a reproducibility experiment from which average yields and standard errors can be obtained. I understand the rationale from the authors about light penetration, Beer-Lambert law etc, but a true scale up also implies accounting for these issues and showing applicability and practicability for industrial developments. Light penetration is often industrially tackled by multiplying irradiation sources, increasing fluence, inserting irradiation systems (light tubes) inside the reaction reactor etc. I am not sure if a scale up is desperately needed, but I feel like the author should acknowledge the limitation of the "numbering-up" approach that they propose.

We thank the Reviewer for their valuable insights and for recognizing the improvements made in our previous revision. We also appreciate the remarks regarding the distinction between "numbering-up" and a scalable sizing-up strategy.

We agree that numbering-up does not address all challenges typically associated with industrial-scale photochemical reactions, particularly those related to photon flux, light penetration, and reactor engineering. Our approach was aimed at demonstrating reproducibility and robustness across multiple microreactors, as a preliminary validation step toward process intensification. However, we have now clarified that our numbering-up strategy should not be interpreted as a comprehensive scale-up, but rather as a practical, modular approach to address throughput enhancement while maintaining reaction performance. We have also included a short discussion of potential scale-up routes, including the integration of internal irradiation systems, multi-source light fields, or intensified flow-through designs, as commonly pursued in industrial photochemistry.

■ Reviewer #4

Some of the previous comments have been addressed, and the revised manuscript shows improvement. However, several important issues remain insufficiently explained. I recommend the authors provide further clarification based on the following comments. I will reconsider the publication of the manuscript once these concerns are fully addressed.

We thank the Reviewer for their engagement with our work and for recognizing the improvements made in the revised version. In response, we have made additional revisions to further refine our manuscript and ensure a more rigorous and transparent presentation of our work.

Comment #1: It remains unclear whether byproduct 4a was formed when the $\text{Ni}_1@n\text{CN}_x$ catalyst was used. According to the authors' own cited reference (Nat. Synth. 2023, 2, 1092–1103), $\text{Ni}_1@n\text{CN}_x$ is expected to favor C–O coupling, which in this case would yield byproduct 4a. The authors must clearly state this and provide a mechanistic rationale for the observed difference in coupling selectivity (C–C vs. C–O) between the current semi-heterogeneous system and the previously reported heterogeneous system. Without such clarification, the mechanistic underpinnings of the semi-heterogeneous system - the core focus of the manuscript - remain ambiguous and must be addressed in the revised manuscript.

While we appreciate the Reviewer's feedback, we respectfully believe that the rationale for the selectivity switch has been clearly demonstrated. The central goal and challenge of this project was to shift the selectivity toward C–C bond formation. This observed difference in selectivity is attributed to a key mechanistic distinction in the decarboxylation step between our system and previously reported conditions. Under our optimized conditions, the decarboxylation step is significantly favored, enabling C–C bond formation. This conclusion is supported by both experimental and computational evidence. Stern–Volmer experiments revealed a markedly stronger interaction between the photocatalyst ($n\text{CN}_x$) and the deprotonated substrate **1** in the presence of Na_2CO_3 compared to Cs_2CO_3 . This enhanced interaction

facilitates the single electron transfer (SET) step that results in the decarboxylation. Furthermore, our computational analysis indicates that Na_2CO_3 promotes decarboxylation more effectively due to a stronger electron-withdrawing interaction between the Na^+ counterion and the carboxylate group.

To further support our hypothesis, we applied the conditions reported in *Nat. Synth.* **2023**, 2, 1092–1103, using DMF as solvent, Cs_2CO_3 or K_2CO_3 as base, and white light irradiation to our semi-heterogeneous system. Under these conditions, we observed exclusive formation of the C–O bond product, with no C–C bond formation. This result is consistent with the findings of MacMillan and co-workers, where also was reported a preference for C–O coupling using DMF as solvent and Cs_2CO_3 as base (see Supporting Information of *Nature* **2016**, 536, 322–325).

In conclusion, the selectivity for C–C vs. C–O bond formation is closely tied to the system's ability to promote the decarboxylation step. In this study, we believe that we have provided both qualitative and quantitative evidence to support the origin of the observed selectivity and to clarify the mechanistic divergence between the two systems. In the revised version, **we have added a few additional sentences to further emphasize this point and to highlight the key factors governing the divergent reactivity.**

Comment #2: While I acknowledge the practical limitations associated with photochemical scale-up, I remain unconvinced that the use of a numbering-up strategy in a batch system sufficiently demonstrates the scalability of the catalytic process. As discussed in reference 43 (*Chem. Rev.* **2022**, 122, 2752–2906), numbering-up is a strategy more appropriately applied to continuous flow systems. In contrast, the current study employs a batch process, where this approach is less relevant. To substantiate claims of scalability, the authors should either demonstrate a genuine scale-up within a batch system or apply a numbering-up strategy in a flow system.

We thank the Reviewer for this thoughtful and important comment regarding scalability. We fully acknowledge that numbering-up is most commonly and effectively applied in continuous flow systems, as noted in reference 43 (*Chem. Rev.* **2022**, 122, 2752–2906). However, in the context of our study, the numbering-up strategy was used as a proof-of-concept to emphasize the critical role of maximizing light-exposed surface area, particularly relevant in heterogeneous systems, where light penetration is further hindered by solid components.

In response to the Reviewer's suggestion, **we have now included a scale-up experiment using a sizing-up approach within the batch system.** The reaction was scaled up by a factor of 10, from 0.2 mmol to 2 mmol, in a single round-bottom flask illuminated with four Kessil lamps (427 nm), yielding product **3a** in 63% isolated yield. This yield is lower than the 77% yield obtained using the numbering-up method. We attribute this 14% drop primarily to differences in the geometry of the reaction vessels, which affected the effective irradiated surface area: in the numbering-up setup, small vials provided an irradiated surface area of approximately $2.16 \text{ cm}^2 \text{ mL}^{-1}$, whereas the round-bottom flask used in the scale-up experiment had a slightly lower irradiated surface area of about $1.57 \text{ cm}^2 \text{ mL}^{-1}$. These results, now detailed in the revised manuscript, help clarify the relationship between reactor geometry, light exposure, and reaction efficiency in batch photochemistry, and demonstrate a ten-fold scale up is feasible with only a modest loss in yield.

Comment #3: The issue regarding yield comparison remains unresolved. The standard conditions in the present study require 72 hours at elevated temperature, whereas MacMillan's protocol achieves C–C coupling in 48 hours at room temperature. Under these markedly different conditions, similar yields (e.g., 71% in this study vs. 85% in MacMillan's work for compound **3a**) do not constitute a valid comparison. Furthermore, the article cited by the authors (*Chem. Sci.* **2019**, 10, 5837–5842) compares mechanochemical and solution-phase conditions within a single study, carefully controlling all variables aside from the activation method. In contrast, the current manuscript compares yields from two independent reports without such control, which invalidates the justification provided. To enable a sound comparison, the authors must present data obtained under matched experimental conditions. Without these corrections, I respectfully maintain that the justification for the manuscript remains unconvincing.

Our intention was not to claim strict equivalence between the two protocols, but rather to highlight that, despite the differing conditions, our method delivers synthetically useful yields using a more accessible setup.

That said, we acknowledge the limitations of comparing results from independent studies and have revised the text to clarify this point, explicitly stating that the comparison is qualitative and does not account for all variables. A direct comparison under identical conditions is, in fact, not feasible due to fundamental differences between the two catalytic systems: MacMillan's protocol employs a homogeneous catalyst, whereas our approach uses a heterogeneous system. These differ not only in terms of active site location and availability, but also in mass and heat transfer properties, dispersion, and reaction dynamics. Attempting to apply identical conditions to both systems would not result in a meaningful comparison, as each catalyst requires a distinct optimization strategy tailored to its physicochemical characteristics.

In response to the Reviewer's suggestion, **we have already a control experiment in which the nCN_x photocatalyst is replaced with a homogeneous Ir-based photocatalyst (Ir[dF(CF₃)ppy]₂(dtbbpy)PF₆) under the otherwise identical conditions we have developed.** This substitution resulted in a 61% yield of product **3a** with MacMillan's catalyst in *Nature* **2016**, 536, 322–325, supporting the relevance and effectiveness of our system (**Table 1, Entry 7**).

Point-by-point response to the Reviewers' comments

(original comments in blue, replies in black, actions in **bold**)

■ Reviewer #4

I have reviewed the revised manuscript and the responses to the reviewers' comments. The major issues have been adequately addressed, and the manuscript is now suitable for acceptance.

We thank the Reviewer for the positive feedback.